



# Measurement and modelling of rainfall partitioning by deciduous *Potentilla fruticosa* shrub on the Qinghai-Tibet Plateau, China

Si-Yi Zhang [a, b], Xiao-Yan Li [b, c]

[a] Guangdong Key Laboratory of Agricultural Environment Pollution Integrated Control, Guangdong Institute of Eco-Environmental Science & Technology, Guangzhou 510650, China

[b] State Key Laboratory of Earth Surface Processes and Resource Ecology, Beijing Normal University, Beijing, 100875, China

[c] College of Resources Science and Technology, Beijing Normal University, Beijing, 100875, China

Correspondence to: Si-Yi Zhang (zdxzqyzsy@163.com)

**Abstract.** Rainfall partitioning is a key process of ecosystem water cycle which has not get enough attention in the alpine deciduous shrub. Moreover,there is no apposite analytical model that suits to estimate the rainfall redistribution of the deciduous shrub with great changes in coverage and leaf storage capacity due to foliation and defoliation. Field and laboratory experiments to assess these processes and to determine vegetation and atmospheric parameters were established for deciduous *Potentilla fruticosa* shrub on the Qinghai-Tibet Plateau. Based on the experimental data, a variable parameter Gash model especially for deciduous shrub rainfall partitioning which is adapted from the revised Gash model was developed to represent these processes. The performance of the variable parameters Gash model in modeling the rainfall partitioning of the deciduous shrub patches is better than that of the original model. The variable parameters Gash model





treated the water captured by the stem as that captured by leaf canopy considering the low height of shrub and its low coverage in leafless period. Thus, the model developed here were perfect to represent the rainfall partitioning processes. It was found that the interception, throughfall and stemflow accounted for 21.44%, 29.26% and 49.30% of gross

rainfall in the shrub patches during the growing season of 2012, respectively. 42.6% of the interception was loss by evaporation from the saturated leaf canopy during rainfall events. The results indicate that adaptations of the analytical model in this paper improved its performance and provide hypotheses more near the realities in the deciduous shrub.

## 1 Introduction

Rainfall partitioning by vegetation is a key process of ecosystem water cycle. The gross precipitation reaches the canopy is redistributed to interception, stemflow and free throughfall (Crockford and Richardson, 2000). These processes have important effect on runoff, soil moisture, geochemical cycle, erosion, and so on (Levia and Frost, 2003; Bryant *et al*., 2005; Llorens and Domingo, 2007; Li *et al*., 2009). A lot attention has been paid to

the rainfall partitioning by different types of plant, forest, shrub, grass and crop in all over the world. Shrub is a common top biological community in arid or semiarid region. Former researches have reported that vegetation in arid or semiarid region has high efficient redistribution of rainfall to adapt to the water limited habitats (Carlyle-Moses, 2004; Owens *et al*., 2006; Johnson and Lehmann, 2009; Garcia-Estringana *et al*., 2010; Li, 2011).



Interception, stemflow and throughfall accounts for 3.0–47.6%, 2.0–43.3% and 31.5–83.3% of the gross precipitation, respectively (Table 1). The interception of canopy evaporates to the atmosphere directly, and this flux is an important composition of ecosystem evapotranspiration and energy fluxes and has great effect on the water and energy balance

at local and catchment scale (Llorens and Domingo, 2007; Zhang, 2014). The stemflow funnels part of rainfall to the ground around the plant trunks or stems, this flux usually is small, accounting for 11.9% of gross precipitation (Table 1), but has important effect on runoff generation, spatial patterning of soil moisture, water erosion, groundwater recharge, geobiochemical circulation and understory ecosystem components (Levia and Frost, 2003).

A lot of models have been developed to estimate rainfall partitioning by vegetation for better understanding their process and composition. (Muzylo *et al*., 2009). Regression model were easy got but unsuitable to adapt to other species or location with different canopy structure and weather condition. Analytical models established by Rutter (Rutter *et al*., 1971; 1975), Gash (Gash, 1979; Gash *et al*., 1995) and Liu (Liu, 1998) were mostly

used in rainfall partitioning researches. These models estimate interception, stemflow and throughfall using a series of parameters containing canopy structures and weather characteristics. These parameters can be got from direct measurement or indirect regression. The models always base on some hypothesis of the conditions and process of wet canopy evaporation, stemflow generation and so on. Mostly, these models commonly

were applied in forests, whereas, application in shrub or grass ecosystems are relatively



rare (Muzylo *et al*., 2009), partly due to the difficulty of water flow measurement techniques (Dunkerley, 2000). The canopy structures of shrub are obvious different from those of forest. Some canopy structure parameters of shrub can be much easier to be directly measured than those of trees because that the shrub is smaller than tall trees.

In most forest researches, the parameters of canopy structure and some weather condition in analytical models are set as constants or semi-constants (Muzylo *et al*., 2009), such as Su *et al*. (2016a). However, the canopy structure often changes seasonally. Especially, the deciduous vegetation canopy changes greatly from leafless to leafed season. Some researchers have change the canopy parameters seasonally (Muzylo *et al*., 2012) or

monthly (Deguchi *et al*., 2006). However, studies focused on rainfall partitioning of changing vegetation canopy are relatively rare, and more studies are required to better represent the components and process of rainfall partitioning influenced by changing vegetation canopy (Muzylo *et al*., 2009; Carlyle-Moses and Gash, 2011). Changing parameters in model according to their changes over time could improve the estimated

results of rainfall partitioning.

      This study aims to measure the rainfall partitioning of deciduous *Potentilla fruticosa* shrub on the Qinghai-Tibet Plateau, and to model their rainfall partitioning using adapted analytical model. The adapted model will consider the special canopy structure of deciduous shrubs. Some hypotheses would be reset due to the special canopy structure and

weather condition. The changes of canopy parameter relating to the process of foliation



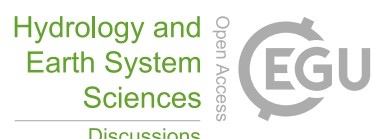

and defoliation were monitored and some important canopy parameter measured directly. The results using adapted model were compared with those of the original model. The specific objectives of this study were (1) to measure and analyze the rainfall interception, stemflow and throughfall of *P. fruticosa* shrub on the Qinghai-Tibet Plateau, China, and (2) to adapt the revised Gash model to the deciduous shrub using the directly measured variable parameters and to compare the results.

## 2 Materials and methods

## 2.1 Study site

The study was conducted at the north of Qinghai Lake watershed, on the northeast of the Qinghai-Tibet Plateau, China (37.594°N, 100.006°E, 3353 m above sea leaf). The site belongs to an alpine cold semiarid region. The mean annual temperature recorded at a National Weather Station 33 km downstream of the experimental site is 0.1 °C and temperature in January is –14.3 to –9.8 °C, and in July is 10.9 to 15.6 °C. The mean annual precipitation is 389.4 mm, and 85% precipitation happens from May to September. The pan evaporation is 1300 to 2000 mm. The dominant species is *P. fruticosa* shrub. It grows as patches with multiple stems connecting with underground stems. The coverage of shrub is 26%, and the height of shrub is 35.4 cm. The mean area of a *P. fruticosa* patch is 0.23 m$^2$. Other accompanying herb species are *Kobresia pygmaea*, *Carex moorcroftii*, *Potentilla saundersiana* and *Polygonum viviparum*. Due to the high elevation and cold



weather, the grow season is short and begins at late May and ends at early October. In this study, only the rainfall partitioning in the *P. fruticosa* shrub patches was considered, that in the inter-patch was not considered.

## 2.2 Measurements of rainfall partitioning and model parameters

During the period 1 June 2012 and 11 September 2012, the rainfall partitioning at the *P. fruticosa* patches were measured. The gross precipitation was measured by a tipping rainfall gauge (ARG100, Campbell, USA). The rainfall events were discretized by assuming without rainfall between events of 12 h, so that there is enough time for the canopy to dry out. 12 h not 8 h (the Gash model recommended, Gash, 1979) was chosen

here because the evaporation rate in the alpine ecosystem is relative lower than other ecosystem in low altitude (Zhang, 2014). The stemflow was measured following the method of Zhang *et al*. (2015a). A small sink was established by wrapping a piece of aluminium foil at the base of stem, the collected stemflow in the sink was drained to a storage bottle through a flexible plastic tube. Because it is very difficult to collect stemflow

in the remote area, we did not measure stemflow for each rainfall events, and we measured and recorded stemflow eight times during the study period, i.e., 11, 20 and 29 June, 17 and 31 July, 22 August and 2 and 11 September, respectively. Totally, 6 stems were selected to measure stemflow. The diameters of the 6 stems ranged from 3.42 to 7.52 mm. The aboveground biomass of the 6 stems was collected when the experiment ended and was

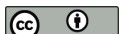



oven-dried at 65 °C and then weighed. A regression equation for the relationship between the stem canopy structure and rainfall characteristic was obtained using stepwise regression method following Zhang *et al*. (2015a). The equation was formed as:

$$SF_v = a*q + b*P_G + c*P_G*M_d \qquad (1)$$

where $SF_v$ is the volume of stemflow (ml) collected from a branch in a special period; $q$ is the number of rainfall events that generate stemflow; $P_G$ is gross precipitation (mm); $M_d$ is the dry aboveground biomass of the branch (g); $a$, $b$ and $c$ are the regression coefficients. The aboveground biomass of *P. fruticosa* patches was collected and oven-dried at the same time. The patch stemflow at each rainfall event was calculated using Eq. (1) with rainfall depth and patch aboveground biomass.

The throughfall was measured using 9 collecting bottles. The throughfall collecting bottles are 3.34 cm in diameter and 15 cm in height with a funnel at the top. The bottles were placed under 3 *P. fruticosa* patches, 3 bottles for each patch. Because the canopy is relatively low, a hole about 10 cm in depth was dug for placing each bottle, and the mouth of bottle was about 5 cm above the ground. The throughfall was recorded at the same time as the stemflow.

The evaporation rate calculated by the energy balance Bowen Ratio method (Zhang, 2014; Zhang et al., 2016a) was used in the rainfall partitioning modeling. The energy balance Bowen Ratio method calculates the evaporation rate using 10 min frequency data of net radiation, soil heat flux, and air temperature and humidity at two different heights,





along with soil moisture and temperature above the layer measuring the soil heat flux. The measurement of original data and calculation process can be referred to Zhang et al. (2016a).

The canopy storage capacity was measured under artificial simulated rainfall. Firstly, the stretch angle of branches of *P. fruticosa* were measured in situ. Then, they were excavated and carefully took back to the laboratory in a whole plant with some soil to assure that they were fresh. In the laboratory, branches were cut off from the base and weighed and then fixed on a wood base at their original angle. Artificial simulated rainfall was implemented immediately. After rainfall, each branch was weighed again, the difference of the weight before and after the rainfall was the water the branch stored. The branch was then disassembled into stems and leaves. The leaves were scanned with a scanner to calculate the one-sided leaf area. The stem and leaves then were oven-dried and weighed. Totally, 33 branches were measured in the simulated rainfall at a rainfall intensity of 10.9 mm h$^{-1}$ and a rainfall duration of 1h. 10.9 mm h$^{-1}$ is the minimum intensity that the rainfall simulator could reach. Through it is much higher than the mean rainfall intensity in the experimental site, 58.6% of the rainfall amount happened in the storms with max 10 min intensity larger than 1.6 mm per 10 min (9.6 mm h$^{-1}$, see the following results in 3.1).

The coverage was estimated by taking and interpreting photos above the canopy in different period. In the leafless period, the coverage is the stem coverage ($c_s$). In the leafed period, the leaf coverage ($c_l$) was interpreted separately. The leaves in nearby shrub patches

were collected at the time of collecting the stemflow and throughfall. The area of the shrub patches was measured and the leaves were scanned to calculate the one-sided leaf area. The leaf area index was calculated as the leaf area of per unit of shrub patch area. The change of the leaf area index was thought to be linear in the foliation and defoliation period.

## 2.3 Description and adapting of revised Gash model

Gash *et al*. (1995) reformulated the original Gash model (Gash, 1975) to better descript the interception of sparse forest. The revised Gash model based on several hypotheses. (1) Rainfall in a given period is segmented to a succession of discrete events, separated by long enough gaps to allow the canopy to dry out after rainfall ceases and interception in each events can be calculated separately. (2) Each rainfall event consists of a wetting up period, a saturation period and a drying out period. (3) The rainfall intensity and the evaporation rates are homogeneous during each rainfall event, and may be considered as constant in all events in the same period. (4) The amount of precipitation lost due to evaporation is related to the coverage of canopy. (5) At the beginning of each rainfall event, the rainfall firstly wets the canopy, and the water captured by trunk is only diverted from canopy after the canopy is saturated. (6) The trunk evaporation only happens in the drying out period. Basing on these hypotheses, several specific canopy and atmospheric parameter are required. The required canopy parameters include the canopy storage capacity ($S$), the trunk storage capacity ($S_t$), the free throughfall coefficient ($p$) and the canopy coverage ($c$),



as well as the stemflow ratio that rainfall is diverted to the trunks ($p_t$). The required atmospheric parameters include incident gross precipitation ($P_G$, mm), mean rainfall intensity ($\bar{R}$, mm h$^{-1}$), the mean evaporation rate ($\bar{E}$, mm h$^{-1}$), the amount of rainfall required to saturate the canopy ($P_G'$) and trunk ($P_G''$). The $P_G'$ and $P_G''$ are calculated with the following equations:

$$P_G' = -\bar{R}S_c / \overline{E_c} \ln(1 - \overline{E_c} / \bar{R}) \qquad (2)$$

$$P_G'' = (\bar{R} / (\bar{R} - \overline{E_c}))(S_t / p_t) + P_G' \qquad (3)$$

where $\overline{E_c}$ is the mean evaporation rate of a wet canopy during a rainfall event, $\overline{E_c} = \bar{E}/c$; $S_c$ is the canopy storage capacity per unit of stand cover area, $S_c = S/c$ (Gash $et\ al.$, 1995). The revised Gash model divides interception into several separate fractions relative to the storms processes (Table 2). The interception ($I$, mm), stemflow ($SF$, mm) and throughfall ($TF$, mm) are calculated using the following equations:

$$\sum_{j=1}^{n+m} I_j = c\sum_{j=1}^{m} P_{Gj} + c\sum_{j=1}^{n}(\overline{E_{cj}} / \overline{R_j})(P_G - P_G') + cnP_G' + cqS_t + cp_t\sum_{j=1}^{n-q}(1 - (\overline{E_{cj}} / \overline{R_j}))(P_G - P_G') \qquad (4)$$

$$\sum_{j=1}^{q} SF_j = cp_t\sum_{j=1}^{n-q}(1 - (\overline{E_{cj}} / \overline{R_j}))(P_G - P_G') - qcS_t \qquad (5)$$

$$\sum_{j=1}^{n+m} TF_j = \sum_{j=1}^{n+m} P_G - \sum_{j=1}^{n+m} I_j - \sum_{j=1}^{q} SF_j \qquad (6)$$

To adapt the revised Gash model to the low deciduous shrub, some hypotheses should be change according to the characteristic of deciduous shrub canopy. The height of the shrub is only about 35.4 cm. The shrub canopy evaporation has no difference from the



stem evaporation and inter-patch herb canopy evaporation. In the leafless and regreening period, the leaves do not cover the stem completely, and rainfall can fall on the stem directly. The canopy of the *P. fruticosa* shrub change greatly along with foliation and defoliation. The aspect change of shrub would have great influence on the canopy coverage

and canopy water storage capacity. To adapt the model, the shrub is distinguished as two part, leaf canopy and stem, and the hypotheses (4) – (6) in the revised Gash model should be change as following: (4) The evaporation from the canopy and from the ground is equal, because the height of the shrub is only about 35.4 cm. So, the amount of precipitation lost due to evaporation is not related to the area of canopy, $\overline{E_c} = \overline{E}$. And the measured

evapotranspiration using the method of BREB (Zhang *et al*., 2016a) could be used as the evaporation in this paper. (5) The coverage of stem in the leafless period is $c_s$ ($0 < c_s < 1$), and the leaf coverage changes daily and is $c_l$ ($0 \leq c_l \leq 1$) at a special period. The rain falls on the stem directly at a probability of $(1 - c_l)c_s$. The rain obtained by the stem also converts to the stemflow at a ratio of $p_t$. When the leaves is saturated, rainfall also diverted to the

stem at a ratio of $p_t$. (6) The water retained on the stems may be treated in a similar manner as that retained by the leaf canopy, as assumed by van Dijk and Bruijnzeel (2001a). The stem evaporation happens in the whole rainfall period but not only in the drying out period. Another hypothesis should be added for calculating the leaf canopy storage: (7) The leaf coverage and leaf canopy storage changes daily and is a function of the leaf dry biomass

or leaf area, as assumed by van Dijk and Bruijnzeel (2001a).





The amount of rainfall required to saturate the leaf canopy ($P_G'$) is calculated with following equation:

$$P_{G,j}' = -S_{l,j}(\overline{R_j}/\overline{E_j})\ln(1-\overline{E_j}/\overline{R_j})$$

(7)

The magnitude of rainfall required to saturate the stem ($P_G''$) is recalculated with the following equation:

$$P_{G,j}'' = (S_s\overline{R_j}/(\overline{R_j}-\overline{E_j})+c_{l,j}P_{G,j}')/(c_s+p_tc_{l,j}-c_sc_{l,j})$$

(8)

The six separate fractions of interception loss in this paper could be recalculated as Table 2, and the interception could be recalculated as the sum of these six fractions. The stemflow could be recalculated by following equation:

$$\sum_{j=1}^{q}SF_j = \sum_{j=1}^{q}p_t(1-\overline{R_j}/\overline{E_j})(c_{l,j}(P_{G,j}-P_{G,j}'')+c_s(1-c_{l,j})(P_{G,j}-P_{G,j}''))$$

(9)

## 3 Results

## 3.1 Meteorological and canopy parameters

During June and October, 2012, there are altogether 80 rainfall events in the experimental site. The total rainfall amounted to 531.0 mm, and ranged from 0.2 mm and 40.0 mm except for a 106.2 mm storm of 50 years' frequency in August. The average rainfall duration and intensity were 11.17 h and 0.9 mm h$^{-1}$. The mean max 10 min rainfall intensity is 0.8 mm per 10 min with a max of 10.4 mm per 10 min, equaling 4.8 mm h$^{-1}$

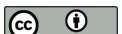

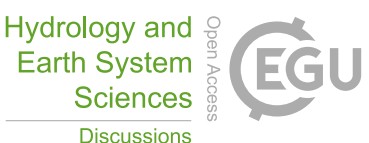

and 624.0 mm h$^{-1}$, respectively. There were 12 rainfall events whose max 10 min intensity were larger than 1.6 mm per 10 min (9.6 mm h$^{-1}$), accounting for 311.4 mm and 58.6% of the total rainfall amount. Most rainfall appeared at July and August, accumulating rainfall depth of 84.4 mm and 248.2 mm, respectively (Figure 1a). Of the 80 rainfall events, 37.5% were less than 0.5 mm, accumulating rainfall depth 6.8 mm; and 11.3 % were 0.5 – 1.7 mm, accumulating rainfall depth 17.0 mm (Figure 1b).

The stem density of the *P. fruticosa* is 385 stems per m$^2$. The average aboveground biomass of the *P. fruticosa* shrub patch in June and August was 1.03±0.39 kg m$^{-2}$. For the *q* events in a period that generated stemflow, the stemflow ($SF_b$, ml) generated by a branch can be calculated as equation:

$$SF_v = -6.26q + 1.31P_G + 0.18P_G * M_d \quad (R^2 = 0.90, \ n = 38) \tag{10}$$

The relationship between the branch water storage capacity ($C_b$, ml) and branch dry biomass ($M_d$, g) is $C_b = 1.10M_d$ ($R^2 = 0.92$, n=33). The relationship between the stem water storage capacity ($C_{st}$, ml) and stem dry mass ($M_{st}$, g) is $C_{st} = 0.60M_{st}$ ($R^2 = 0.95$, n=33) and the relationship between leaves water storage capacity ($C_{lf}$, ml) and leaves dry mass ($M_{lf}$, g) was $C_{lf} = 4.29M_{lf}$ ($R^2 = 0.92$, n=33). The percentage of water storage capacity ratio of stem and leaves was 48.2% and 51.8%, respectively. The shrub patch water storage capacity was 1.13±0.43 mm, and for the leaf and the stem was 0.59 mm and 0.55 mm, respectively.



## 3.2 Observed rainfall partitioning pattern

During June 1 and September 11, 2012, seven measurements contained 55 rainfall events and 484.8 mm rainfall were available in the experiment. The cumulative throughfall, stemflow and interception loss of the seven recorded periods equaled 141.9 mm, 239.0

5   mm, and 103.9 mm, respectively, representing 29.26%, 49.30%, and 21.44% of gross rainfall. The linear relationship between stand stemflow ($SF_{st}$) and gross rainfall ($P_G$) was $SF_{st} = 0.68P_G–2.61q$ ($R^2=0.99$, $n=7$). The $p_t = 0.68$.

## 3.3 Performance of the revised Gash model and the variable parameters Gash model

10   The estimated rainfall partitioning patterns using the revised Gash model are shown in Table 4. The estimated throughfall, stemflow and interception were 124.7 mm, 246.2 mm and 113.9 mm, respectively. In the whole monitoring period, the throughfall was underestimated by 12.07%, and the stemflow and interception were overestimated by 3.01% and 9.6 %, respectively. For a special record, the estimated error was –41.2% ~ –6.5%, –

15   5.69% ~ 30.86% and –20.0% ~ 27.6 % for the throughfall, stemflow and interception, respectively. According to the classified level of models (Muzylo *et al*., 2009), the revised Gash model in this paper was good (5% < error < 10%) for interception, very good (1% < error < 5%) for stemflow, and fair (10% < error < 30%) for the throughfall.

The estimated rainfall partitioning pattern using the variable parameters Gash model



for deciduous shrub is presented in Table 5. Compared to the former results (Table 4), the performance of the new model are much better. The total estimated errors are only –1.2%, –0.8% and 3.5% for the throughfall, stemflow and interception, respectively. According to the classified level of models (Muzylo *et al*., 2009), the model got in this paper was very

good (1% < error < 5%) for throughfall, stemflow and interception.

Rainfall partitioning predicted by the revised Gash model and the variable parameters Gash model were shown in Table 6. The difference of estimated interception by two models was only 6.4 mm or 1.3% of gross precipitation. But the contribution of interception predicted by two models has greatly difference in the $I_s$ and $I_{st}$. The $I_{st}$

predicted by the variable parameters Gash model is 10.2 mm more than that predicted by the revised Gash model, which accounting for 9.8% of the total observed interception. Meanwhile, the $I_s$ predicted by the variable parameters Gash model is 15.0 mm lower than that predicted by the revised Gash model, which occupying 14.4% of the total observed interception.

## 3.4 Parameter sensitivities

Sensitivity analyses of canopy and meteorological parameters were conducted to determine their influence degree on the rainfall partitioning pattern (Figure 2). $c$, $\bar{E}$, $S_c$, $S_t$ and $p_t$ had positive relationship with interception, whereas $\bar{R}$ had negative relationship with interception. The three most sensitive parameters of interception were $\bar{R}$, $c$ and $\bar{E}$. If



the $\bar{R}$ decreased by 50%, the interception would increase by 46.95%; and if the $c$ and $\bar{E}$ decreased by 50%, the interception would decrease by 40.13% and 27.02%, respectively. $c$, $\bar{R}$ and $p_t$ were positively correlated with estimated interception loss, whereas, $\bar{E}$, $S_c$ and $S_t$ were negatively correlated with stemflow yield.

The three most sensitive parameters of stemflow yield were $p_t$, $c$ and $\bar{R}$ when the parameters decreased. If the $p_t$, $c$ and $\bar{R}$ decrease by 50%, the stemflow would decrease by 53.74%, 54.18% and 17.29%, respectively. When the parameters increase, the stemflow is mostly sensitive to the change of $p_t$, $\bar{E}$ and $\bar{R}$. If the $p_t$ and $\bar{R}$ increase by 50%, the stemflow would increase by 54.33% and 6.65%, respectively; and if the $\bar{E}$ increased by 50%, the stemflow would decrease by 8.95%.

$c$, $\bar{E}$, $S_c$, $S_t$ and $p_t$ were negatively correlated with estimated throughfall, whereas $\bar{R}$ were positively correlated with estimated throughfall. The three most sensitive parameters of throughfall were $c$, $p_t$ and $\bar{R}$ if the parameters decrease. If the $c$ and $p_t$ decrease by 50%, the throughfall would increase by 156.06% and 111.81%, respectively. If the parameters increase, the throughfall is most sensitive to the change of $p_t$, $\bar{E}$ and $\bar{R}$. If the $p_t$ and $\bar{E}$ increase by 50%, the throughfall would decrease by 111.81% and 8.76%, respectively; and if the $\bar{R}$ increase by 50%, the throughfall would increase 6.27%.

Therefore, the rainfall partitioning pattern in the shrub patch is very sensitive to the change of canopy coverage, evaporation, rainfall intensity and stemflow ratio. The accurate measurement or estimate of these parameters has great influence on the



effectiveness on the model.

# 4 Discussion

## 4.1 Parameters

The parameters sensitivities of interception from the revised Gash model for interception has been analyzed by Limousin *et al.* (2008), Sun *et al.* (2014), Su *et al.* (2016a). The results in this paper are in accordance with their results that the atmospheric $\bar{E}$ and $\bar{R}$ and canopy parameter $c$ and $S$ are the most sensitive factors of rainfall partitioning modeling. It would not be discussed here. The parameters sensitivities of the stemflow and throughfall were not reported formerly. The stemflow was sensitive to the $p_t$, $c$, $\bar{E}$ and $\bar{R}$. Because the stemflow is the section of interception that runs down the stem, the parameters that affect the interception also affect the stemflow. The proportion $p_t$ also has important in the stemflow of course, it determines the percentage of interception that converts to stemflow. The throughfall is the residue of gross precipitation minus the interception and stemflow, and it is mostly sensitive to the parameter that the interception and stemflow is most sensitive to.

Due to the parameters sensitivities of the revised Gash model, it is necessary to set different parameters at shorter periods according to the variation of vegetation and rainfall (Jackson, 2000; Deguchi *et al.*, 2006; Herbst *et al.*, 2008; Sraj *et al.*, 2008; Muzylo *et al.*, 2012), especially for which interception and stemflow is impressionable. Deciduous





vegetation has great changes in canopy with leafed and leafless period and their transition within a year. Their canopy parameters such as coverage, leaf canopy storage capacity, etc., should not be treated as constants in the growing season due to foliation and defoliation. Especially, the interception, stemflow and throughfall were very sensitive to the canopy coverage.

Canopy coverage is a very important parameter in the analytical model, and many model parameters (e.g. $\overline{E_c}$, $S_c$, $p$, $pt$) have linear relationship with the canopy coverage (Gash *et al*., 1995; Limousin *et al*., 2008). The reduction of the canopy coverage would decrease the interception (Limousin *et al*., 2008). This is quite direct results because when the canopy coverage decrease, the areas capture the rainfall was reduced, and then the interception decreased. As the sensitive analysis, the decrease of coverage could result in the reduction of stemflow and the increase of free throughfall. The stemflow are the part of interception that run down the stem, so if the interception reduces, the stemflow would reduce. The coverage reduce meant the increased of the free throughfall coefficient, which could be assumed to be one minus coverage (Su *et al*., 2016b).The coverage can be estimated using hemispherical photographs taken by a fish eye lens camera or a plant canopy analyzer (Mcjannet *et al*., 2007; Limousin *et al*., 2008; Muzylo *et al*., 2012). If the coverage is not directly measured, it can be assumed to be one minus free throughfall coefficient (Shi *et al*., 2010; Fan *et al*., 2014). The free throughfall coefficient can be estimated as the slope of the linear regression of throughfall against gross precipitation for



small rainfall events that were insufficient to exceed canopy storage capacity (Jackson, 1975; Shi *et al*., 2010; Fan *et al*., 2014). However, these two methods are not suitable for the low deciduous shrub. The height of the shrub in this study is too low to use the fish eye lens camera or the plant canopy analyzer. Because the coverage and canopy storage

capacity changed continuously along with the foliation and defoliation, it was difficult to collect enough throughfall data to get the regression equation for a special period, especially in the arid and semiarid region with limited rainfall events. So, basal coverage of the stem in the dormant period and of the max coverage in leaf luxuriating period were estimated using photographs. And the coverage in transition period was link to the leaf

area index. This method could get the accurate coverage in the whole research period.

The canopy storage capacity is the most important parameter in the interception modeling (Aston, 1979). The determination of the canopy water storage capacity has great influence on the accuracy of the rainfall partitioning modeling (Limousin *et al*., 2008). Precise measurement of the canopy water storage capacity is the precondition of the

prediction accuracy. Several methods were found to estimate the canopy water storage capacity, as the negative intercept of the regression of gross precipitation against the sum of throughfall and stemflow (Wallace and Mcjannet, 2006), or as the negative intercept divided by one minus drainage partitioning coefficient ($p_d$) of the linear regression of throughfall against gross precipitation, fitted with a pre-established slope of ($1-cp_d$)

(Valente *et al*., 1997; Muzylo *et al*., 2012). Some study emphasized that rainfall for



calculating the canopy storage capacity should be saturating events (Klaassen *et al*., 1998; Carlyle-Moses and Price, 1999; Limousin *et al*., 2008). Some researchers also measured the canopy storage capacity directly (Wang *et al.*, 2006; Wang and Zhang, 2006). Direct measurement of canopy capacity was an all-consuming task and was not common used in

forest rainfall partitioning modeling. For the shrub, it could be easier to measure the canopy capacity directly because the magnitude of the shrub is much smaller than the forest. In this paper, the water storage capacity was measured under artificial simulated rainfall. As the rainfall intensity has influence on the canopy storage capacity (Wang *et al*., 2012), the simulated rainfall method is much authentic than the immersion method used by Wang

and Zhang (2006) which could not consider the influence of rainfall intensity. The simulated rainfall method treated the shrub in a close to the real condition, the storage capacity got in this method should be much true. The simulated rainfall is more effective at wetting by formatting coherent droplets (Beysens *et al*., 1991). The water storage capacities measured by immersion were lower than the values obtained in the simulated

rainfall experiments (Garcia-Estringana *et al*., 2010). Canopy storage capacity was assumed to be linearly related to leaf area index, and can be expressed as the product of specific leaf storage and leaf area index (Liu, 1998; van Dijk and Bruijnzeel, 2001a). So, by testing the specific leaf storage capacity in laboratory and monitoring the leaf area index or leaf biomass in different period, the canopy storage capacity can be estimated in a

special period.



The stem storage capacity might be somewhat less important than the canopy storage capacity. In most reported study, the stem storage capacity was much smaller than the canopy storage capacity (Carlyle-Moses and Price, 2007; Muzylo *et al*., 2012). The stem storage capacity could be estimated as the negative intercept of the linear regression of the stemflow against gross precipitation (Su *et al*., 2016a). In this paper, the stem storage capacity was measured along with the leaf canopy storage capacity under simulated rainfall. The leaf canopy storage capacity and stem storage capacity were 0.59 mm and 0.55 mm, respectively. Unlike former reports, the leaf canopy storage capacity of the *P. fruticosa* shrub was only a little larger than the stem storage capacity. There were no former reports about the water storage capacity of the *P. fruticosa* shrub. The specific leaf storage capacity of *A. ordosica*, *H. scoparium* and *S. psammophila* were 0.68, 0.44 and 0.31 g g$^{-1}$ dry weight, respectively (Yang, 2010), and they were 0.51, 0.41 and 0.73 g g$^{-1}$ of specific canopy storage capacity for *C. korshinskii*, *H. scoparium*, and *A. ordosica*, respectively (Wang *et al*., 2012), and 0.49－0.94 mm and 0.16－0.78 mm for the *C. vulgaris* and *P. aquilinum*, respectively (Leyton *et al*., 1967; Hall, 1985; Pitman, 1989). The specific leaf storage capacity of the *P. fruticosa* shrub was higher than those reported results, but fell into the range reported by Garcia-Estringana *et al*. (2010). The reason could be that in this paper, the specific storage capacity of leaf and stem were considered separately, the specific storage capacity of stem was similar as the former reported results of shrubs, whereas, the specific storage capacity of leaf was higher than reported results, because the

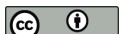



leaf had larger specific surface area per dry mass. The storage capacity of stem was almost as high as that of leaf may be explained by the special bark of the *P. fruticosa*. The bark of this shrub is thick, papery and soft (Figure 3b), and can absorb a lot of water. Such a bark may be the mechanism the shrub adapting the cold weather. The leaf area has great influence on the canopy coverage, canopy water storage capacity and therefore on the interception loss, higher leaf area would result in more interception loss (Gómez *et al*., 2001; van Dijk and Bruijnzeel, 2001a; Deguchi *et al*., 2006). So, it is important to monitor the process of canopy structure changes.

The revised Gash model is sensitive to rainfall intensity and evaporation rate. This sensitivity does not differ between seasons regardless of the structural changes in vegetation (Muzylo *et al*., 2012). The rainfall intensity is often observed by auto-record rainfall gauge and it has no subjective factor. The evaporation has different calculation method and could be a possible source of error in modeling rainfall partitioning (Llorens, 1997). The evaporation could be estimated by the Penman-Monteith equation (P-M equation) (Rutter *et al*., 1971; Carlyle-Moses and Gash, 2011), or estimated as the slope of the regression of interception against gross precipitation (Pereira *et al*., 2009), or was obtained from optimization procedure where evaporation was adjusted to minimize the root mean square difference between the daily modelled and measured interception (Wallace and Mcjannet, 2008). When using the P-M equation, the accurate determination of the aerodynamic resistance is a great difficult, while the evaporation is sensitive to the



aerodynamic resistance (van Dijk *et al*., 2015). The aerodynamic component of evaporation is typically larger than the radiation component (van Dijk *et al*., 2015). Evaporation estimated by the P-M equation was thought to be the main weakness of the Gash models (Zhang *et al*., 2006), and advanced measurement methods should be used to

estimate the evaporation rate in the revised Gash model (Su *et al*., 2016a). Eddy-covariance technique has been regarded as the best method to directly measure evapotranspiration, and was used in the revised Gash model by Zhang *et al*. (2006), but its measurements during rainfall were questionable (Wang and Dickinson, 2012).The Bowen ratio system is a reliable and oecumenical method to monitor the evaporation (Wang and

Dickinson, 2012). In this paper, the evaporation data of energy balance Bowen ratio method from Zhang (2014), Zhang *et al*. (2016a) was used in modeling rainfall partitioning. The average evapotranspiration in the *P. fruticosa* shrub meadow was 0.11 mm h$^{-1}$ during the experimental period. It was relatively lower compared to worldwide evaporation rate calculated by revised Gash model, more than 87% papers reported evaporation higher than

0.11 mm h$^{-1}$ (Murakami, 2007). This is partly relative to the low vegetation height and high-altitude cold weather with low temperature. The hourly evaporation varied greatly in different time, ranging from –0.04 to 0.87 mm h$^{-1}$, controlled mainly by radiation (Zhang, 2014; Zhang *et al*., 2016a). The variation of the evaporation and its importance imply that it should not be set as a constant. With the event-based evaporation, the results of rainfall

partitioning modeling could be improved.



The variable parameters Gash model is relatively more complex than the original models, but has better performance in more complex conditions. The input data demands are more than the revised Gash model, but are not hard to collect at present. The changes in coverage and leaf canopy storage depending on the seasonal aspects of deciduous shrub, as well as rain water captured by the stem directly and evaporated during the whole rainfall period has been included in and is considered a vital conceptual improvement. These improvements are of great significance in deciduous shrub stands which have a great change in coverage and leaf storage capacity relative to their leaf area index which is determined by the foliation and defoliation. These additions along with the consideration for the stem water capturing and evaporation have resulted in the interception, stemflow and throughfall simulations all being very good. The introduction of new parameters and process to the model might violate the principle of "requisite simplicity" for any model, but considering the conceptual clarity and scientific rigor, the new approach adopted in the paper could be meaningful. Extending the variable parameters Gash model to other ecosystems with different canopy structure and weather condition will show whether it works.

## 4.2 Rainfall partitioning in deciduous shrubs

The rainfall fell on the *P. fruticosa* canopy was mainly redistributed to stemflow, accounting for 49.3%. In most researches, the main part of rainfall partitioning is




throughfall (Llorens and Domingo, 2007). The stemflow took the main part in the *P. fruticosa* partly because the research space scale is based on the shrub patch, and the inter-patch is not considered. In the individual shrub scale, Belmonte Serrato and Romero Diaz (1998) also reported an as high as 43.3% of relative stemflow production. Another reason is the stem density of the *P. fruticosa* is as high as 385 stems per $m^2$, and its canopy structures that promote stemflow yield. The stem was nearly vertical and rarely warped above the ground, and their sub stems have small angles (Figure 3a).

There is no former report of the stemflow measurement of the P. *fruticosa*, whereas there were two researches on the interception of the P. *fruticosa*. 30.56% and 10.42% of gross precipitation was intercepted by the shrub canopy as reported by Nie (2009), Li (2015). The relative interception found in the present study is 21.4% and is among the former observed range. This value is common among the former reports (Table 1).

$I_s$ and $I_a$ calculated from the revised Gash model took 53.4% and 20.1% of interception of the *P. fruticosa* shrub. It is common that the $I_s$ and $I_a$ is the main part of the interception calculated from the revised Gash model. In the 32 former results, the mean $I_s$ 42.8%, ranging from 9.0% to 91.2%, and the mean $I_a$ is 36.7%, ranging from 4.0% to 74.0% (Table 7). These results also reveal why the interception is sensitive to the $\bar{E}$ and $S_c$.

The $I_{st}$ calculated from the revised Gash model took 17.5% of the interception. This value is much higher than the average value in the 32 former results, 4.5% of the interception (Table 7). The reason lies in the high stem storage capacity of the *P. fruticosa*.



The stem storage capacity (0.55 mm) is almost as high as the leaf canopy storage capacity (0.59 mm).

The most difference of the compositions of the interception between the variable parameters Gash model and the revised Gash model lies in the $I_s$ and $I_{st}$ (Table 6). The $I_s$ calculated by the variable parameters Gash model is smaller than that calculated by the revised Gash model, and the $I_{st}$ calculated by the variable parameters Gash model is larger than that calculated by the revised Gash model. This is partly because the variable parameters Gash model assumed that the water captured by the stem was treated as that captured by the canopy. In the revised Gash model, water captured by the stem is all from the canopy interception and only evaporated in the drying out period after the end of rainfall. In the variable parameters model, rain could drop on the stem directly, and water captured by stems could evaporate in and after the rainfall. We thought the improved model is more near the actual condition, especially, in the leafless period when the stem was exposed to the atmosphere completely. The assumption in the variable parameters model increased income and the evaporation time of the water captured by stem, thus the $I_{st}$ increased.

## 5 Conclusion

This study also confirms that rainfall partitioning by the canopy plays a significant role in ecosystem hydrological cycle. 21.44% of gross rainfall was intercepted by the canopy,

29.26% and 49.30% of gross rainfall was available water that free fell and drained along the stem into the soil during the growing season of 2012. The interception proportion was similar to the values reported for other arid and semiarid shrubs. However, while the stemflow was higher than former reported values, partly because their high stem density and canopy structures that promote stemflow yield.

The variable parameters Gash model adapted for the deciduous shrub is presented with hypotheses more in accordance with the real condition and has better performance than the revised Gash model. The revised Gash model underestimated the throughfall by 12.07%, and overestimated the stemflow and interception by 3.01% and 9.6 %, respectively. Whereas, the variable parameters Gash model performed all very well in the prediction of throughfall, stemflow and interception, with only –1.2%, –0.8% and 3.5% estimated errors, respectively.

The wet leaf canopy evaporation during storms accounted for 42.6% of total interception loss, implying the importance of the evaporation rate in rainfall partitioning modelling. The interception loss via stem evaporation estimated by variable parameters Gash model is much higher than the proportion estimated by the revised Gash model. This proportion is the most difference between these two models. The results reveal the important of the stem evaporation in the interception loss of the deciduous shrubs.

**Acknowledgements**



This work was supported by the National Science Foundation of China (Grant No. NSFC 41501295, 41130640 and 91425301), SPICC Program, projects from State Key Laboratory of Earth Surface Processes and Resource Ecology.

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

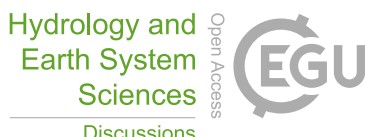
## Tables:

Table 1 Summary of relevant researches on rainfall partitioning of shrubs in arid and semi-arid regions, the current paper is added for completeness. MAP is mean annual precipitation. $P_G$ is gross precipitation.

| References | Sites | MAP (mm) | Shrub species | Function group | Stemflow % | Interception % | Throughfall % | Scale |
|---|---|---|---|---|---|---|---|---|
| 1. Serrato Romero (1998) Belmonte and Diaz | Europe Lomo Herrero, Spain | 600 | *J. oxycedrus* | evergreen | 18.7 | 36.5 | 44.5 | individual |
| | | | *R. officinalis* | evergreen | 43.3 | 25.2 | 31.5 | individual |
| | | | *T. vulgaris* | evergreen | 29.8 | 33 | 37.2 | individual |
| 2. Domingo et al. (1998) | Rambla Honda, Spain | 300 | *A. cytisoides* | deciduous | 20 | 40 | 40 | patch |
| | | | *R. sphaerocarpa* | deciduous | 7 | 21 | 72 | patch |
| 3. Garcia-Estringana et al. (2010) | Mediterranean region, Europe | N/A | *M. strasseri* | evergreen | 3.8 | | | individual |
| | | | *D. pentaphyllum* | evergreen | 21 | | | individual |
| | | | *C. arborescens* | deciduous | 17.3 | | | individual |
| | | | *R. sphaerocarpa* | deciduous | 9.7 | | | individual |
| | | | *C. ladanifer* | evergreen | 5.6 | | | individual |
| | | | *C. albidus* | evergreen | 20.8 | | | individual |
| | | | *R. officinalis* | evergreen | 23.2 | | | individual |
| | | | *L. latifolia* | evergreen | 25.7 | | | individual |
| | American | | *C. zeelandia* | evergreen | 26.4 | | | individual |



| Reference | Location | (mm) | Species | Leaf habit | | | | Scale |
|---|---|---|---|---|---|---|---|---|
| 4. Návar and Bryan (1990) | Linares, Mexico | 740 | A. farnesiana | deciduous | 3 | 27.2 | 69.8 | stand |
| | | | D. texana | deciduous | 10.5 | | 60 | — |
| | | | P. laevigata | evergreen | | | | — |
| 5. Mauchamp and Janeau (1993) | Chihuahuan desert, Mexico | 283 | F. cernua | deciduous | 28.6 | 10.5 | 53.4 | — |
| 6. Martinez-Meza and Whitford (1996) | Chihuahuan desert, USA | 230 | F. cernua | deciduous | 10.6 | 36 | 53.4 | — |
| | | | P. glandulosa | deciduous | 5.4 | 31.1 | 63.5 | — |
| | | | L. Tridentata | evergreen | 10 | 34.2 | 55.8 | — |
| 7. Carlyle-Moses (2004) | Sierra Madre Oriental, Mexico | 635 | Matorral Community | evergreen | 8.5 | 8.2 | 83.3 | stand |
| 8. Owens et al. (2006) | Edwards Plateau, USA | 600–900 | J. ashei | evergreen | 5 | 40 | 55 | stand |
| 9. Li et al. (2008) | Loess Plateau, China | 263 | T. ramosissima | deciduous | 2.2 | | | individual |
| | | | C. korshinskii | deciduous | 7.2 | | | individual |
| 10. Li et al. (2009) | Mu Us desert, China | 395 | R. soongorica | deciduous | 3.7 | | | individual |
| | | | H. scoparium | deciduous | 3.4 | | | individual |
| | | | S. psammophila | deciduous | 6.3 | | | individual |
| 11. Wang et al. (2013) | Tengger desert, China | 190 | C. korshinskii | deciduous | 7.2 | | | individual |
| | | | A. ordosica | deciduous | 2 | | | individual |
| 12. Jian et al. (2014) | Loess Plateau, China | 420 | C. korshinskii | deciduous | 12.3 | | | individual |
| | | | H. rhamnoides | deciduous | 8.4 | | | individual |



Hydrology and Earth System Sciences — Discussions — Open Access

| Reference | Location | Rainfall | Species | Leaf habit | | | | Scale |
|---|---|---|---|---|---|---|---|---|
| 13. Zhang et al. (2015a) | Qinghai-Tibet Plateau, China | 378 | *M. squamosa* | deciduous | 2.3–10.2 | | | stand |
| 14. Yuan et al. (2016) | Transitional zone between Loess Plateau and Mu Us desert, China | 437 | *S. psammophila* | deciduous | 5.54 | | | individual |
| 15. Zhang et al. (2015b) | Shapotou Desert, China | 191 | *C. korshinskii* | deciduous | 9 | 16.7 | 74.3 | individual |
| | | | *A. ordosica* | deciduous | 2.9 | 22.3 | 74.8 | individual |
| 16. Wang et al. (2011) | Shapotou Desert, China | 191 | *C. korshinskii* | deciduous | 8 | | | individual |
| 17. Zhang et al. (2016b) | Shapotou Desert, China | 191 | *C. korshinskii* | deciduous | | 29.1 | | individual |
| | | | *A. ordosica* | deciduous | | 17.1 | | individual |
| 18. Li et al. (2016) | Mu Us sandy Desert, China | 345 | *A. ordosica* | deciduous | 8.56 | | | individual |
| 19. Peng et al. (2014) | Typical steppe of Inner Mongolia, China | 392 | *C. microphalla Lain* | deciduous | 3.11 | 20.86 | 76.04 | patch |
| 20. Yue et al. (2013) | Horqin Sandy Land, Northeast China | 351.7 | *S. gordejevii* | deciduous | 2.19 | 15.03 | 82.78 | individual |
| 21. Ma et al. (2012) | Qinghai-Tibet Plateau, China | 378 | *M. squamosa* | deciduous | 4.04 | 47.56 | 48.4 | individual |
| 22. Li (2015) | Heihe River watershed, China | 400–600 | *D. fruticosa* | deciduous | | 30.56 | | patch |
| | | | *C. jubata* | deciduous | | 22.22 | | patch |



| Reference | Location | Precip. | Species | Leaf | | | | Type |
|---|---|---|---|---|---|---|---|---|
| 23. Xu et al. (2013) | Badan Jilin desert, China | 115.9 | H. ammodendron | deciduous | | 16.6 | | patch |
| | | | T. ramosissima | deciduous | 33.1 | | | patch |
| 24. Jian et al. (2012) | Loess Plateau, China. | 427 | N. tangutorum | deciduous | | 12 | | individual |
| | | | C. korshinskii | deciduous | | 12.3 | 59.7 | individual |
| | | | H. rhamnoides | deciduous | 18.3 | 8.4 | 73.3 | individual |
| 25. Yang et al. (2008) | Mu Us sandy Desert, China | 395 | S. psammophila | deciduous | 2.9 | 24.9 | 72.2 | individual |
| 26. Wang et al. (2016) | Tengger Desert, China | 191 | C. korshinskii | deciduous | 11.3 | 14.3 | 74.4 | individual |
| | | | A. ordosica | deciduous | 5.5 | 32.7 | 61.8 | individual |
| 27. Nie (2009) | Qilian Mountains, China | 300~600 | B. diaphana | deciduous | | 22.46 | | patch |
| | | | S. gilashanica | deciduous | | 20.48 | | patch |
| | | | C. jubata | deciduous | | 16.76 | | patch |
| | | | P. fruticosa | deciduous | | 10.42 | | patch |
| | | | C. tangutica | deciduous | | 6.96 | | patch |

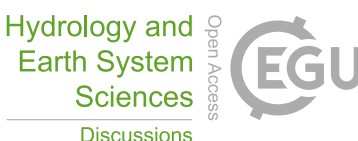

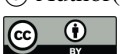

Table 2 Fractions of interception in the revised Gash model and adapted model in this paper.

| Fractions | Variable | Determination | Analytical form Revised Gash model | Variable parameter Gash model (this paper) |
|---|---|---|---|---|
| I | | For $m$ storms insufficient to saturate the leaf canopy ($P_G < P'_G$) | | |
| 1 | $I_c$ | Evaporation from the whole canopy | $c\sum_{j=1}^{m} P_{G,j}$ | $\sum_{j=1}^{m} c_{t,j} P_{G,j}$ |
| II | | For $n$ storms sufficient to saturate the leaf canopy ($P_G \geq P'_G$) | | |
| 2 | $I_w$ | Wetting the leaf canopy | $cnP'_G - cnS_c$ | $\sum_{j=1}^{n} c_{t,j} P'_{G,j} - S_{t,j}$ |
| 3 | $I_s$ | Wet leaf canopy evaporation during storms | $c\sum_{j=1}^{n}(\overline{E_c}/\overline{R})(P_{G,j}-P'_G)$ | $\sum_{j=1}^{n} c_{t,j}(\overline{E_j}/\overline{R_j})(P_{G,j}-P'_{G,j})$ |
| 4 | $I_a$ | Evaporation after storms cease | $cnS_c$ | $\sum_{j=1}^{n} S_{t,j}$ |
| 5 | $I_{st}$ Part i | Evaporation from saturate the stem in $q$ storms sufficient to stem | $cp_t\sum_{j=1}^{n-q}(1-(\overline{E_c}/\overline{R}))(P_{G,j}-P'_G)$ | $\sum_{j=1}^{n-q}(1-\overline{E_j}/\overline{R_j})(c_{t,j}p_t(P_{G,j}-P'_{G,j})+c_s(1-c_{t,j})P_{G,j})$ |
| 6 | $I_{st}$ Part ii | Evaporation from stem in $n-q$ storms insufficient to saturate the stem | $cqS_t$ | $qS_{st}+\sum_{j=1}^{q}\overline{E_j}c_s(1-c_{t,j})(P_{G,j}-P''_{G,j})/\overline{R_j}$ |



Table 3 Observed rainfall partitioning pattern in the *P. fruticosa* shrub patches

| Record date | Rainfall events | | Rainfall | | Throughfall | Stemflow | Interception |
| | Total | *q* | Gross | *q* events | | | |
| | | | mm | mm | mm | mm | mm |
| 2012/6/11 | 7 | 3 | 22.8 | 20.20 | 6.35 | 7.83 | 8.6 |
| 2012/6/20 | 7 | 6 | 40.8 | 40.60 | 15.45 | 12.37 | 13.0 |
| 2012/6/29 | 7 | 6 | 67.4 | 67.20 | 21.80 | 28.80 | 16.8 |
| 2012/7/17 | Data missing | | | | | | |
| 2012/7/31 | 9 | 6 | 70.8 | 68.40 | 19.50 | 30.93 | 20.4 |
| 2012/8/22 | 15 | 7 | 206.8 | 204.80 | 60.04 | 121.46 | 25.3 |
| 2012/9/2 | 5 | 3 | 42.2 | 41.80 | 9.50 | 20.41 | 12.3 |
| 2012/9/11 | 5 | 2 | 34.0 | 33.40 | 9.20 | 17.20 | 7.6 |
| Total | 55.0 | 33.0 | 484.8 | 476.4 | 141.9 | 239.0 | 103.9 |





Table 4 Estimated rainfall partitioning pattern in the *P. fruticosa* shrub patches using the revised Gash model. EV: Estimated values, EE: Estimated error

| Record Date | Throughfall | | Stemflow | | Interception | |
|---|---|---|---|---|---|---|
| | EV | EE | EV | EE | EV | EE |
| | mm | % | mm | % | mm | % |
| 2012/6/11 | 4.95 | −22.12 | 8.41 | 7.39 | 9.4 | 9.6 |
| 2012/6/20 | 9.09 | −41.2 | 16.19 | 30.86 | 15.5 | 19.6 |
| 2012/6/29 | 18.26 | −16.22 | 35.69 | 23.91 | 13.5 | −20 |
| 2012/7/17 | Data missing | | | | | |
| 2012/7/31 | 16.47 | −15.57 | 31.59 | 2.13 | 22.7 | 11.7 |
| 2012/8/22 | 57.08 | −4.94 | 117.44 | −3.31 | 32.3 | 27.6 |
| 2012/9/2 | 10.12 | 6.5 | 19.25 | −5.69 | 12.8 | 4.4 |
| 2012/9/11 | 8.77 | −4.67 | 17.63 | 2.46 | 7.6 | 0.1 |
| Total | 124.7 | −12.07 | 246.2 | 3.01 | 113.9 | 9.6 |





Table 5 Estimated rainfall partitioning pattern in the *P. fruticosa* shrub patches using the variable parameters Gash model

| Record Date | Throughfall | | Stemflow | | Interception | |
|---|---|---|---|---|---|---|
| | EV | EE | EV | EE | EV | EE |
| | mm | % | mm | % | mm | % |
| 2012/6/11 | 7.0 | 10.6 | 8.6 | 9.9 | 7.2 | ‑16.8 |
| 2012/6/20 | 12.1 | ‑21.8 | 16.1 | 30.4 | 12.6 | ‑2.9 |
| 2012/6/29 | 22.6 | 3.7 | 33.1 | 14.8 | 11.7 | ‑30.3 |
| 2012/7/17 | Data missing | | | | | |
| 2012/7/31 | 18.0 | ‑7.6 | 29.5 | ‑4.6 | 23.3 | 14.3 |
| 2012/8/22 | 58.3 | ‑2.8 | 115.2 | ‑5.2 | 33.3 | 31.6 |
| 2012/9/2 | 11.7 | 23.6 | 17.8 | ‑12.6 | 12.6 | 2.6 |
| 2012/9/11 | 10.4 | 12.6 | 16.7 | ‑2.7 | 6.9 | ‑9.1 |
| Total | 140.2 | ‑1.2 | 237.1 | ‑0.8 | 107.5 | 3.5 |





Table 6 Rainfall partitioning predicted by the revised Gash model and the new Gash model. RG: value predicted from the revised Gash model, VG: value predicted from the variable parameters Gash model.

| Items | | Amount | | Percentage of gross precipitation | | Percentage of total Interception | |
|---|---|---|---|---|---|---|---|
| | | RG | VG | RG | VG | RG | VG |
| | | mm | mm | % | % | % | % |
| Throughfall | | 124.7 | 140.2 | 25.7 | 28.9 | | |
| Stemflow | | 246.2 | 237.1 | 50.8 | 48.9 | | |
| Interception | Total | 113.9 | 107.5 | 23.5 | 22.2 | 100 | 100 |
| | $I_c$ | 7.4 | 6.9 | 1.5 | 1.4 | 6.5 | 6.4 |
| | $I_w$ | 2.8 | 2.9 | 0.6 | 0.6 | 2.5 | 2.7 |
| | $I_s$ | 60.8 | 45.8 | 12.5 | 9.5 | 53.4 | 42.6 |
| | $I_a$ | 22.9 | 21.9 | 4.7 | 4.5 | 20.1 | 20.4 |
| | $I_{st}$ | 19.9 | 30.1 | 4.1 | 6.2 | 17.5 | 28.0 |





Table 7 Summary of components of simulated interception using revised Gash model

| Reference | Sites | MAP[a] | Species | $I$ mm | $I_c$ % | $I_w$ % | $I_s$ % | $I_a$ % | $I_t$ % |
|---|---|---|---|---|---|---|---|---|---|
| 1. Dykes (1997) | Temburong District, Brunei | 4582 | Mixed Dipterocarp rainforest | 68.4 | 4.7 | 1.9 | 63.0 | 27.8 | 2.6 |
| 2. Carlyle-Moses and Price (1999) | Ontario, Canada | 785 | *Q. ruba*,*A. saccharum*,*F. grandifolia* | 41.3 | 4.8 | 2.2 | 27.1 | 59.8 | 5.8 |
| 3. Aboal *et al*. (1999) | Tenerife, Spain | 733 | laurel forest | 166.0 | 12.0 | 1.0 | 9.0 | 74.0 | 3.0 |
| 4. van Dijk and Bruijnzeel (2001b) | Java, Indonesia | 2600 | *M. esculenta* Crantz; *Z. mays* L.; *O. sativa* L. | 311.5 | 0.0 | 4.6 | 91.2 | 4.0 | 0.2 |
| | | | | 132.8 | 0.1 | 8.1 | 84.0 | 7.6 | 0.3 |
| 5. Link *et al*. (2004) | Washington, USA | 2467 | *P. menzesii*, *T. heterophylla*, and *T. plicata* | 126.7 | 29.0 | 4.0 | 30.0 | 37.0 | – |
| | | | | 106.9 | 35.0 | 5.0 | 17.0 | 44.0 | – |
| | | | | 136.9 | 40.0 | 5.0 | 11.0 | 44.0 | – |
| 6. Deguchi *et al*. (2006) | Aichi Prefecture, Japan | 1498.4 | *Q. serrata* and *C. barbinervis,* etc. | 699.52 | 4.8 | 3.2 | 70.8 | 29.3 | – |
| 7. Zhang *et al*. (2006) | Hunan province, China | 1200–1500 | *C. japonica, N. indicum, E. japonicas, T. gymnanthera* | 118.5 | 3.3 | 12.9 | 32.0 | 48.1 | 3.7 |
| 8. Murakami (2007) | Japan | 1467.7 | *C. obtusa*, 1999 | 320.1 | 2.0 | 1.1 | 80.1 | 8.9 | 7.9 |
| | | | *C. obtusa*, 2000 | 256.4 | 2.4 | 1.3 | 73.9 | 11.9 | 10.4 |
| 9. Wallace and Mcjannet (2008) | Queensland, Australian | – | Oliver Creek | 1085.0 | 14.7 | 2.4 | 21.5 | 58.9 | 2.6 |
| | | | Hutchinson Creek | 900.0 | 13.8 | 2.8 | 34.6 | 45.8 | 3.1 |
| | | | Mount Lewis | 1193.0 | 21.3 | 3.8 | 25.6 | 47.3 | 2.0 |
| | | | Mount Lewis | 509.0 | 22.6 | 4.1 | 25.9 | 44.8 | 2.6 |
| | | | Upper Barron | 428.0 | 15.4 | 4.0 | 46.0 | 32.7 | 1.9 |
| 10. Limousin *et al*. (2008) | Montpellier, France | 908 | *Q. ilex*, control plot | 396.5 | 31.2 | 2.8 | 17.8 | 38.1 | 10.1 |
| | | | *Q. ilex*, thinned plot | 260.1 | 31.5 | 4.9 | 17.9 | 35.8 | 9.9 |
| 11. Shi *et al*. (2010) | Liupan Mountains, China | 591.6 | *P. armandii* | 72.5 | 1.7 | 2.7 | 39.5 | 55.2 | 0.9 |
| | | 1405 | *B. aemula* R.Br. | 158.6 | 0.0 | 2.5 | 77.3 | 16.6 | 3.6 |

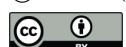



| 12. Fan *et al.* (2014) | Bribie Island, Australia | | *P. elliottii* Engelm, *P. caribaea* Morelet var. *hondurensis* | 211.6 | 1.4 | 1.9 | 51.6 | 34.8 | 7.4 |
|---|---|---|---|---|---|---|---|---|---|
| 13. Sun *et al.* (2015) | Mt. Karasawa, Japan | 1265 | *C. obtusa*, pre-thinning | 209.3 | 4.4 | 2.5 | 62.9 | 26.8 | 3.4 |
| | | | *C. obtusa*, post-thinning | 104.4 | 2.3 | 2.8 | 45.4 | 40.3 | 9.2 |





**Figures:**

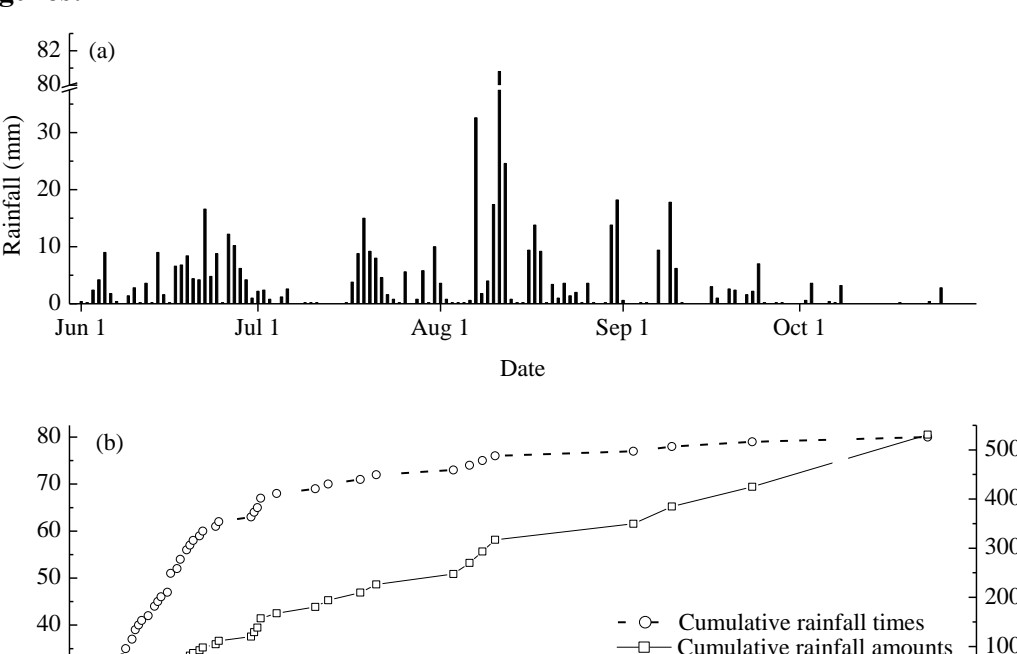

Figure 1 (a) daily rainfall depth during June and October, 2012; (b) cumulative rainfall events and amount of different individual event rainfall depth.





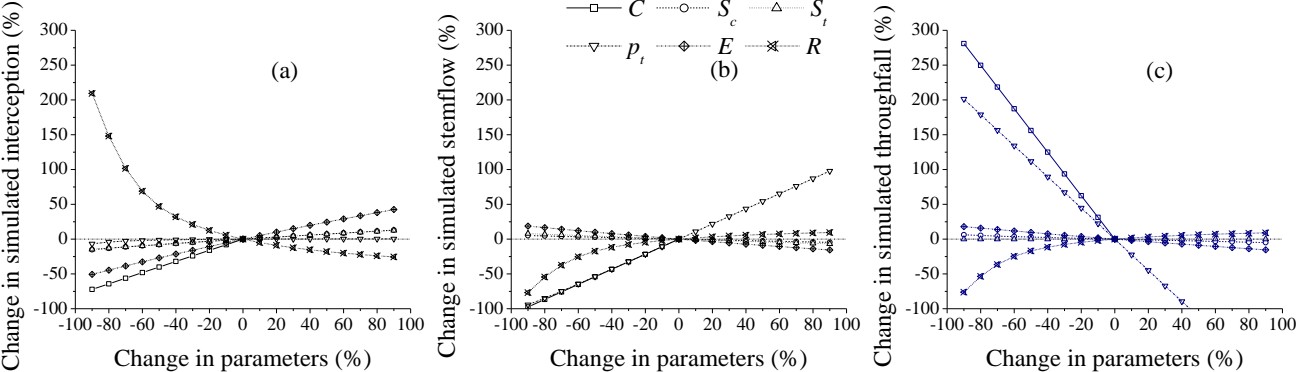

Figure 2 Sensitivity analysis of the parameters of the revised Gash model





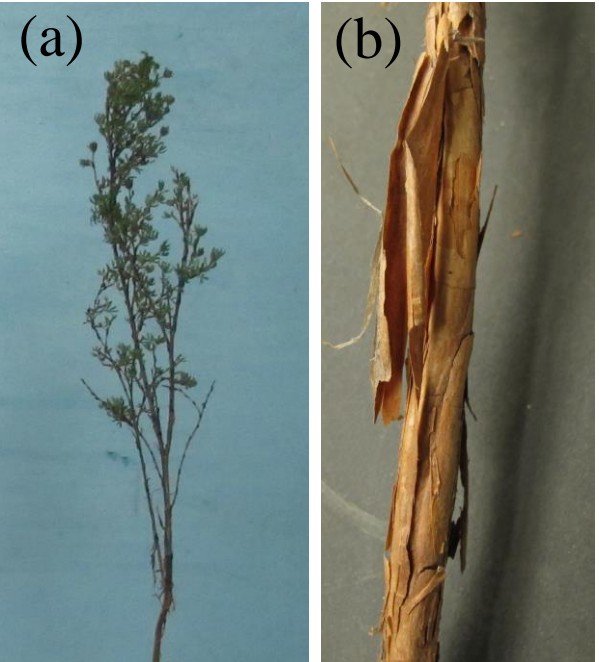

Figure 3 Photos of (a) a typical *P. fruticosa* branch and (b) its bark