# Peer review of "Measurement and modelling of rainfall partitioning by deciduous *Potentilla fruticosa* shrub on the Qinghai-Tibet Plateau, China"

_Hydrology and Earth System Sciences, 2016_

## Referee Comment (RC1) · J. Van Stan (Referee) · 17 Dec 2016

Manuscript #2016-589 by Zhang and Li examines/models rainfall partitioning of a shrub species in an alpine semiarid site (Qinghai Lake, China). Although there are some interesting aspects (i.e., directly measured variable canopy structural parameters in the model), my opinion is that this study does not require publication in an international top-tier journal outlet. Thus, I recommend rejection from HESS. Rather, I believe this study is better suited for a journal outlet focused on the region within which it is situated as substantial (and excellent) work has already been published on the precipitation partitioning of plant canopies in this area (the authors' works cited section includes many examples). Yes, this study adds one more shrub species to the list of plants

studied (with some details on canopy structural variability), but is that broad enough?

Besides this issue, I have other concerns:

1) Methods are missing details. Specifically...

**Was only 1 rainfall gauge used? For the past several years, rainfall measurement protocols have necessitated three rainfall gauges (i.e., see International Co-operative Programme – ICP Forests). If only 1 rainfall gauge was used, please justify and acknowledge the difference between this study and current standard rainfall measurement protocols.

**How were the few throughfall gauges distributed in the patches? Seeing as very few throughfall gauges were deployed (see point 2) and throughfall is spatially heterogeneous, knowledge of the arrangement of gauges is necessary to provide the reader an idea of how well represented the spatial heterogeneity was in the study's observations. Since there were so few gauges, were they roved around? Other concerns about throughfall observations shared later (see point 2)

**What were the dimensions of the stemflow collection devices? It is mentioned that "sinks" of aluminum foil (P6, L12) were used to collect stemflow. How big was the sink area? Sinks that extend far from the shrub stem may be gathering throughfall as well as stemflow. This might explain stemflow accounting for nearly 30% of gross rainfall – a rather high, albeit possible, proportion. Knowing the dimensions of the stemflow "sink" will strengthen (or weaken) confidence in the quite large stemflow production. Also, how long after a storm event were stemflow gauges manually measured? If stemflow measurements were not taken immediately after a storm, were there efforts to minimize evaporation losses from the collectors? Other concerns about stemflow observations shared later (see point 3)

2) Throughfall variability may have been too under-sampled. There were very few throughfall collectors (n = 9 total, n = 3 per plot: P7, L11-14) each with a small collection area (3.34 cm diameter: P76, L12), which likely prevents accurate throughfall estimation considering the well-documented spatial variability of throughfall.

3) Stemflow estimates may not be representative due to selective sampling. Stemflow observations were selected from very few storms (n = 8: P6, L16). Did these storms represent the continuum of storm magnitudes and intensities generally experienced at the site? If not, meteorological conditions that favor stemflow generation may explain the high stemflow proportion. Stemflow observations were also selected from very few stems (n = 6: P6, L17). Where were these stems located in the patches? Were they on the edge or interior to the shrub patch? Were the patched trimmed to install stemflow collars (which may create an artificial edge effect)? This is important to know as location within the patch can affect stemflow generation. Also, how did the selected stems, and canopy draining to those stems, compare to the range of canopy characteristics at the site?

4) I have some minor concerns with the edits to the reformulated Gash model in this study. I say "minor" because it is regarding only 2 assumptions that simplified evaporation estimates:

(a) The authors' assume that "shrub canopy evaporation has no difference from the stem evaporation" (P10, L18 – P11, L1) but provide no data in support of the assumption. Without data supporting the authors' claim, I'm inclined to believe that stem and canopy evaporation rates would be different due to reasons commonly identified in past literature: (i) canopy shading the stem, (ii) different albedo of leaf and stem surfaces, (iii) wind speeds being reduced from canopy edge to the interior stem, and (iv) complex stem bark surfaces (like shown in Fig. 3b) may shelter entrained water from meteorological conditions driving evaporation.

(b) The authors' also assume that "evaporation from the canopy and from the ground is equal, because the height of the shrub is only about 35 cm"; however, the physical drivers of evaporation can differ between the ground and shrub canopy despite modest

differences in elevation.

5) The manuscript is in need of significant English language editing. As it would take too much time to identify and suggest changes for all of the necessary language editing, an example in each section are provided to guide the authors during their revisions:

Abstract, P1, L10– "has not get enough attention" should be "has not gotten enough attention". . . but, my opinion is that the language shouldn't be so colloquial. It would be better to state something like "has not received enough attention"

Introduction P2, L11– "The gross precipitation reaches the canopy. . ." should be "The gross precipitation that reaches the canopy. . ." and the authors incorrectly state that the canopy partitions precipitation only into interception, stemflow and "free" throughfall. This ignores all "release" throughfall produced from canopy contact. Why not just say "interception stemflow and throughfall" as the general term "throughfall" implies the sum of free and release throughfall?

Methods P6, L7– "rainfall events were discretized by assuming without rainfall between events of 12 h. . ." should be something like "rainfall events were discretized by assuming a minimum inter-event time of 12h. . ." Minimum inter-event time is a common term in precipitation partitioning literature (i.e., Dunkerley, 2015, Hydrol Process, 29, 3294 and Llorens et al., 2014, J Hydrol, 512, 254). I would also recommend the authors' follow the convention for introducing species: latin name (taxonomic authority, common name). Thereafter, the use of the abbreviated latin name is typically used.

Results P15, L13 – I think the line "which occupying 14.4% of the total observed interception" should be "which accounted for 14.4% of the total observed interception"

Discussion P17, L11-12 – "The proportion pt also has important in the stemflow of course" needs to be rewritten for clarity as I'm unsure what the authors are saying.

Conclusion P26, L9 to P27, L2 – this statement is unclear. Does "available water that free fell and drained along the stem" mean "stemflow"? Why would "free" falling

droplets "drain along the stem"? Or, does this statement simply mean "throughfall and stemflow"? Please revise for clarity.

---

## Referee Comment (RC2) · Anonymous Referee #2 · 3 Mar 2017

**General comments:**

The manuscript (ms) reports on measurement and modelling of rainfall interception by a deciduous shrub species. Although several studies have already been published on the rainfall interception by deciduous shrubs, only in a few modelling was done. The specific characteristics of these cover-types, with drastic seasonal changes in canopy structure, could make this study quite useful and liable to provide relevant contributions on the subject. However, I think that the ms has several important shortcomings in the present form and that its focus/rationale needs to be improved and clarified. In my opinion, the ms needs a major revision before it can be considered for publication in HESS.

[Figure]

**Specific comments:**

1. The English is poor and the ms does not read well (sometimes it is hard to understand what the authors are trying to say).

2. In some cases, standard terminology on rainfall interception is not used correctly by the authors. Usually, "interception" is used to describe the interaction process between rainfall and vegetation while "interception loss" refers to its evaporation component (the amount of water retained in plants surfaces that evaporates back into the atmosphere). The authors use the term "interception" with both meanings resulting in a confusing text (e.g., page 2, line 10−12, "The gross precipitation reaches the canopy is redistributed to interception, stemflow and free throughfall"; page 18, lines 12−14, "The stemflow are the part of interception that run down the stem, so if the interception reduces, the stemflow would reduce"). The authors should check all text and differentiate between concepts using the appropriate terminology.

3. The description of the experimental site and vegetation characteristics needs further information and to be reorganized. In page 5, lines 16−17, the authors say "The coverage of shrub is 26%, and the height of shrub is 35.4 cm". How was this cover fraction evaluated? Does this value correspondent to the (average?) cover fraction of an individual plant or to the total percentage of cover area in the experimental site? Is the given value for shrub height a mean? What about other characteristics of individual plants (average number of stems per plant, mean diameter of each stem,...)? Although some of this data is presented in the ms, it is dispersed across several sub-sections (e.g., page 13, line 7). All this disperse information should be gathered together.

4. Concerning the measurement of rainfall, throughfall, stemflow and micrometeorological variables, important information is missing. Location and type of the different gauges (tipping bucket and/or bottles) are not given. How were rain gauge locations chosen? How far from the edge of the patches were they placed? Did the gauges/bottles stay in fixed positions or were moved to new random positions each time they were measured? At what height were the micrometeorological sensors installed? Where were these sensors installed: above a shrub patch or in open areas between patches? What is the footprint for these data? Although micrometeorological data is from a previous study, it should be briefly described here. All this information is relevant to the study (measurement and modelling of rainfall interception) and should be presented in the ms.

An aerial photography of the site with the location of the used devices (rainfall, throughfall and stemflow gauges and the Bowen ratio tower), would be helpful.

5. To extrapolate stemflow measurements to the total patch area the authors used a stepwise methodology to derive a regression model. Which were the independent variables considered in this analysis? Though the final model has only three variables (page 7, eq. 1), were other structural features/rainfall characteristic considered?

    One of the variables included in eq. 1 is $q$, "the number of rainfall events that generate stemflow" (page 7, lines $5-6$). How was $q$ evaluated? In page 6, lines $14-16$, it is stated that "Because it is very difficult to collect stemflow in the remote area, we did not measure stemflow for each rainfall events, and we measured and recorded stemflow eight times during the study period". Given this, how do the authors know the number of rainfall events that generate stemflow in each period?

6. It seems to me that the authors do not totally understand the sparse version of Gash's analytical model.

    (a) They say that the model requires several parameters and refer that "the free

throughfall coefficient ($p$) and the canopy coverage ($c$)" are two of them (page 9, line 19). In page 18, lines 16−17, they re-state that $p$ is a parameter of the model. This is not correct! The sparse version of the Gash model only requires $c$, the proportion of covered area relative to the total area.

(b) Although not acknowledged, the authors mix the sparse version of the Gash model proposed by Gash et al. (1995) with the slightly different version presented later by Valente et al. (1997) (e.g., the amount of rainfall required to saturate the trunks ($P_g$") is only defined by Valente et al. (1997)).

(c) Two of the most important parameters of the sparse version of the Gash model are $\overline{R}$ and $\overline{E}_c$ (and not $\overline{E}$, as it is said in page 10, line 3). According to Gash et al. (1995), these parameters are the mean rainfall rate and the mean evaporation rate during saturated conditions, respectively, and should be representative for the whole modelling period. Following Gash (1979), the method usually used to derive $\overline{R}$ is the average of all hours with rainfall equal or greater than 0.5 mm (two bucket tips) for the whole modelling period. How did the authors calculate $\overline{R}$? Nothing is said about this. The same happens with $\overline{E}_c$. The authors say they used data obtained with the Bowen Ratio/Energy Balance method (BREB) (page 11, lines 10−11), but do not say how.

(d) Besides, it seems that the authors do not fully understand the meaning of $\overline{E}_c$. It represents the evaporation rate at which intercepted water can evaporate from a fully saturated canopy. But the authors say that $\overline{E}_c = \overline{E}/c$ (page 10, line 8). What is the meaning of $\overline{E}$ in the context of the sparse version of the model? If $\overline{E}$ is the actual measured evaporation rate from a fully wet vegetation and it is assumed that the only water source is the studied wet vegetation then this relationship is correct. Otherwise, it is not. It seems to me that the authors did not get it correctly. In fact, the authors say (page 23, lines 12−13) that "the average evapotranspiration in *P. fruticosa* shrub meadow was 0.11 mm h$^{-1}$ during the experimental period". How was this

calculated? They also refer that "the hourly evaporation varied greatly in different time, ranging from $-0.04$ to $0.87$ mm h$^{-1}$, controlled mainly by radiation" (page 23, lines $16-17$). However, during rainy/cloudy conditions (when the canopy is saturated), radiation is typically low and evaporation rate should not change much. This may suggest that the aforementioned values include periods where the vegetation is not fully wet, possibly not representative of saturated canopy conditions.

(e) The authors present three equations (page 10) to calculate the different components of rainfall interception (interception loss, stemflow and through-fall). Although based in the model version proposed by Valente et al. (1997) (again not acknowledge here), these equations do not describe the sparse version of the Gash model. As the authors say (page 9, lines $11-13$), one of the assumptions of the model is that $\overline{E}_c$ and $\overline{R}$ are assumed constant over the whole modelling period. However, while gross precipitation seems to be constant (since the $j$ index is missing in $P_g$), but should not, $\overline{E}_c$ and $\overline{R}$ can change from storm to storm (because they have a $j$ index). Moreover and contrary to the current practice, trunk storage capacity ($S_t$) is expressed in mm on a projected cover area basis (that is why it is necessary to multiply $S_t$ by $c$ in eq. 4 and 5). Whenever the units of a parameter are water depth (e.g., mm), it should be clearly stated in the text what is the reference area (e.g., ground area, covered area, ...).

(f) The authors present a new version of this model to adapt it to the studied deciduous shrub (page 10, line 17 to page 12, line 10). They assume that the evaporation rates from all the vegetation components (canopy, stems and inter-patch herbs) are the same. I am not sure if this is a realist assumption, since roughness and/or the micrometeorological conditions are seldom similar. Nevertheless, the requirements of the energy and water balances should be met. When all the vegetation is saturated, the measured BREB values ($\overline{E}$) represent the evaporation of the total area and not just of the

wet shrubs cover (see my previous comment 6.(d)). It seems to me that the authors did not took into account the water balance equation in their new modelling proposal (page 12, eqs. 8 and 9 and Table 2). How were these new equations obtained? An explanation is needed.

(g) Another important missing information is the "time-step" used to run the model. Although the model is storm-based, it is usually run assuming that each rainday is an independent rainfall event. Which procedure did the authors used?

7. The authors present results on the water storage capacity of leaves and stems (page 13, lines 12−16) but do not explain how they were obtained. Only the method used to measure branch water storage capacity is described. Furthermore, they do not explain how ml were converted into mm (page 13, line 18). What is the reference area in the latter?

The method used to estimate another model parameter ($p_t$) is not also described in the text.

8. Considering the characteristics of the studied vegetation (deciduous), it would be expectable the presentation of data on the time evolution of some parameters, namely canopy cover, and canopy and stem storage capacities. This would provide support on the need of using time variable parameters instead of the usual constant values. Besides, as the authors used different $\overline{E}_c$ and $\overline{R}$, it would be relevant to have a graph of their values along the modelling period. Neither of these variable parameters, nor the constant ones needed to run the sparse version of the Gash model are given in the ms.

9. The performance of the tested models were only evaluated by the total error (EE). However, EE per se does not evaluate the quality of model performance throughout the simulation period. For that purpose, authors should have applied

some additional measure, such as modelling efficiency (see Mayer and Butler, 1993, Ecol. Modelling, 68: 21-32).

10. As in many other studies, the authors have conducted a sensibility analysis of the sparse version of the Gash model. The question is: what is new about this? If they have used their own model version this could be interesting. What has been done is just a repetition that does not bring any new insight on the subject. Furthermore, the presentation of the results and their discussion are incomplete. Why is not shown a positive change of c in the graphs (Fig. 2)?

In what concerns canopy cover ($S$), model sensitivity to this parameter was found to be very small which is not in accordance with most previous findings. However, the authors state that "the results in this paper are in accordance with [the] results" of other studies and will not be discussed in the ms (page 17, lines $6-8$). On the other hand, they state that "the canopy storage capacity is the most important parameter in the interception modelling" (page 19, lines $11-12$) which is contradictory. In my opinion, the authors should focus their work in what is new and relevant to the subject (modelling the rainfall interception process in a deciduous shrub cover).

11. **Minor comments:**

   (a) Page 3, line 13 & page 4, line 6 $-$ replace "Analytical" by "Conceptual". The Rutter model is not an "analytical model".

   (b) Page 4, line 6 $-$ what are semi-constants?

   (c) Page 4, lines $16-20$ & page 4, lines $1-6$ $-$ the objectives of the work should be presented in a concise way. This text should be simplified and avoid repetitions.

   (d) Page 6, line 6 $-$ specify tip sensitivity of rainfall gauge.

(e) Page 6, lines 16−17 − there are only seven periods with measurements. Data from the 17th July 2012 is missing (Tables 3, 4 and 5). Authors should mention that in the text.

(f) Page 6, line 18 − are stem diameter units correct (mm)? A stem with 3.4 mm seems too small to support any collecting device to measure stemflow.

(g) Page 7, line 19 − what is the meaning of "10 min frequency data"? Do the authors mean "10 min average data"?

(h) Page 8, lines 15−17 − this sentence should go to the discussion section.

(i) Page 9, line 6 − replace (Gash, 1975) by (Gash, 1979).

(j) Page 11, line 10 − the acronym BREB should be previously defined.

(k) Page 12, lines 1 and 4 − the subscript $j$ is missing in the symbols.

(l) Page 12, line 15 − do the authors mean a storm with 50 years' return period?

(m) Page 13, line 9 − according to eqs. 1 and 10, symbol for stemflow should be $SF_v$, not $SF_b$.

(n) Table 1 − please remove the reference to $P_g$; this variable is not in table.

(o) Table 3 − table not referred in text.

(p) Figure 1 b) − I do not understand this graph. What do the authors want to show with it? Please explain.

---

## Author Comment (AC1) · 4 Apr 2017

Dear J. Van Stan,

We would like to thank you for your valuable and constructive comments. The comments are very helpful to the improvement of the manuscript, and will be well incorporated into the revision of the paper. The following paragraphs respond to your comments one by one.

General comments:

Manuscript #2016-589 by Zhang and Li examines/models rainfall partitioning of a shrub species in an alpine semiarid site (Qinghai Lake, China). Although there are some

interesting aspects (i.e., directly measured variable canopy structural parameters in the model), my opinion is that this study does not require publication in an international top-tier journal outlet. Thus, I recommend rejection from HESS. Rather, I believe this study is better suited for a journal outlet focused on the region within which it is situated as substantial (and excellent) work has already been published on the precipitation partitioning of plant canopies in this area (the authors' works cited section includes many examples). Yes, this study adds one more shrub species to the list of plants studied (with some details on canopy structural variability), but is that broad enough? Besides this issue, I have other concerns:

Thanks! This study not only adds one more shrub species to the list of plants studied, what is more important, this study adapts the revised Gash model according to the seasonal change of canopy structures of deciduous shrubs, which is rarely reported before. The new version model performs better than the original model and can be used in other deciduous ecosystem.

The modelling of rainfall partitioning on a deciduous shrub is rare. Although there are a lot of work has already been published on the precipitation partitioning of shrubs, the modelling of precipitation partitioning of shrub is not as common as that of forests (Muzylo et al., 2009), partly due to the difficulty of water flow measurement techniques for shrubs (David, 2010). The published models are firstly developed for forest. The canopy structures of shrubs are obvious different from those of forests. Some canopy structure parameters of shrub can be much easier to be directly measured than those of trees because that the shrub is smaller than tall trees. What is more, in most published researches, the model parameters of canopy structure and some weather condition are set as constants (Muzylo et al., 2009). However, the canopy structure often changes seasonally. Especially, the deciduous vegetation canopy changes greatly from leafless to leafed season. This paper focused on the influence of the characteristic of a short shrub canopy and its seasonal changing on the rainfall partitioning. Deciduous shrubs are common top biological communities in arid or semiarid region, not limited in the

Qinghai-Tibet Plateau, China. The new version of model can be applied to similar ecosystem with deciduous shrubs.

1) Methods are missing details. Specifically... 2) **Was only 1 rainfall gauge used? For the past several years, rainfall measurement protocols have necessitated three rainfall gauges (i.e., see International Cooperative Programme – ICP Forests). If only 1 rainfall gauge was used, please justify and acknowledge the difference between this study and current standard rainfall measurement protocols.

Yes, only one rainfall gauge was used. The same as the technical recommendations of ICP Forests, the rainfall gauge was located in a relatively flat, open area, and about 1 m above the ground. The canopy had no influence on the rainfall gauge as its height is much lower than the that of the rainfall gauge. It is common to use only one or two rainfall gauges in local gross rainfall measurement in rainfall partitioning researches according to what I know (eg: Muzylo et al., 2012; Návar, 2013; Macinnis-Ng et al., 2014). And I do not find that the rainfall measurement protocols have necessitated three rainfall gauges in the downloaded Meteorological Measurements MANUAL from http://www.icp-forests.org/pdf/manual/2016/Manual_Part_IX.pdf.

**How were the few throughfall gauges distributed in the patches? Seeing as very few throughfall gauges were deployed (see point 2) and throughfall is spatially heteroge-neous, knowledge of the arrangement of gauges is necessary to provide the reader an idea of how well represented the spatial heterogeneity was in the study's observa-tions. Since there were so few gauges, were they roved around? Other concerns about throughfall observations shared later (see point 2)

Thanks! The shrub patches are relatively homogeneous with dense and short stems. It is difficult to deployed too many and too large throughfall gauges, because larger gauges might stick on the stems and collect the stemflow. We had put more throughfall gauges in the patches in the experiment, but we found some throughfall was anoma-lous later, those stuck on stems had collected much more water, even more than the

gross precipitation, maybe some from the stemflow. These conditions are very different from those of forests. The anomalous throughfall was not included in our analysis. We also found the spatial heterogeneity of throughfall in a patch is little. We deployed six gauges in the patches cores and three within the boundary.

**What were the dimensions of the stemflow collection devices? It is mentioned that "sinks" of aluminum foil (P6, L12) were used to collect stemflow. How big was the sink area? Sinks that extend far from the shrub stem may be gathering throughfall as well as stemflow. This might explain stemflow accounting for nearly 30% of gross rainfall – a rather high, albeit possible, proportion. Knowing the dimensions of the stemflow "sink" will strengthen (or weaken) confidence in the quite large stemflow production. Also, how long after a storm event were stemflow gauges manually measured? If stemflow measurements were not taken immediately after a storm, were there efforts to minimize evaporation losses from the collectors? Other concerns about stemflow observations shared later (see point 3)

Thanks! We made the sinks as small as possible as the rain intensity in the region is small. The sinks edges were about $1\sim 2$ mm away from the wrapped shrub stems (Figure 1). The throughfall fell in the sinks was less than 1.5% of the stemflow and it was ignored. The stemflow gauges were nearly sealed with one small hole 5 mm in diameter connecting the plastic tube for the stemflow draining in and one small hole about 3 mm in diameter for air discharge. The air discharge hole was opened for air pressure balance when the stemflow flowed in. The evaporation losses from the collectors through the two small holes was little and could be ignored.

Figure 1 Stemflow collection apparatus on a branch (photo courtesy of Si-Yi Zhang)

3) Throughfall variability may have been too under-sampled. There were very few throughfall collectors (n = 9 total, n = 3 per plot: P7, L11-14) each with a small collection area (3.34 cm diameter: P76, L12), which likely prevents accurate throughfall estimation considering the well-documented spatial variability of throughfall. 4) Thanks! The

shrub patches are relatively homogeneous with dense and short stems. It is difficult to deployed too many and too large throughfall gauges, because larger gauges might stick on the stems and collect the stemflow. We had put more throughfall gauges in the patches, but we found those stuck on stems had collected much more water, even more than the gross precipitation, maybe some from the stemflow. These conditions are very different from those of forests. The anomalous throughfall was not included in our analysis. We also found the spatial heterogeneity of throughfall in a patch is little. We deployed six gauges in the patches cores and three within the patches boundary.

5) Stemflow estimates may not be representative due to selective sampling. Stemflow observations were selected from very few storms (n = 8: P6, L16). Did these storms represent the continuum of storm magnitudes and intensities generally experienced at the site? If not, meteorological conditions that favor stemflow generation may explain the high stemflow proportion. Stemflow observations were also selected from very few stems (n = 6: P6, L17). Where were these stems located in the patches? Were they on the edge or interior to the shrub patch? Were the patched trimmed to install stemflow collars (which may create an artificial edge effect)? This is important to know as location within the patch can affect stemflow generation. Also, how did the selected stems, and canopy draining to those stems, compare to the range of canopy characteristics at the site? 6) Stemflow observations was measured eight times during the study period, i.e., 11, 20 and 29 June, 17 and 31 July, 22 August and 2 and 11 September, respectively. Unfortunately, some stemflow and throughfall data was missed in July 17, 2012. The other seven periods included 55 storms (See Table 3 in the original ms, q: The number of rains which generated stemflow) ranging from 0.2 mm to 106.2 mm. The average rainfall duration and intensity were 11.17 h and 0.9 mm h–1. The mean max 10 min rainfall intensity is 0.8 mm per 10 min with a max of 10.4 mm per 10 min, equaling 4.8 mm h-1 and 624.0 mm h-1, respectively. There were 12 rainfall events whose max 10 min intensity were larger than 1.6 mm per 10 min (9.6 mm h-1), accounting for 311.4 mm and 58.6% of the total rainfall amount. These storms can represent the continuum of storm magnitudes and intensities generally experienced at the site.

[Figure]

Four of the six stems were interior to the shrub patch and other two were on the edge. The patched were not trimmed to install stemflow collars. The typical stems were selected representing different diameters, stem length, coverage and location.

7) I have some minor concerns with the edits to the reformulated Gash model in this study. I say "minor" because it is regarding only 2 assumptions that simplified evaporation estimates: 8) (a) The authors' assume that "shrub canopy evaporation has no difference from the stem evaporation" (P10, L18 – P11, L1) but provide no data in support of the assumption. Without data supporting the authors' claim, I'm inclined to believe that stem and canopy evaporation rates would be different due to reasons commonly identified in past literature: (i) canopy shading the stem, (ii) different albedo of leaf and stem sur- faces, (iii) wind speeds being reduced from canopy edge to the interior stem, and (iv) complex stem bark surfaces (like shown in Fig. 3b) may shelter entrained water from meteorological conditions driving evaporation. (b) Thanks. We agree to the fact that there is difference between the stem and canopy evaporation rates due to the reasons the reviewer identified. We proposed this assume just think it could be better than the original assume that the trunk evaporation only happens in the drying out period. The original assume of course is not the real fact, too. Especially, in the leafless period, rain can fall on the stems and then evaporated directly. The shading of the leaf canopy was considered, and the evaporation rate of the wet stems is , where cl,j and cs is the leaf and stem coverage, respectively, and is the evaporation rate measured by Bowen ratio and energy balance method. The evaporation is a complicated process, and it is difficult to distinguish evaporation rates on different surfaces. The original and present assumes both are for simplify in modeling these process.

(c) The authors' also assume that "evaporation from the canopy and from the ground is equal, because the height of the shrub is only about 35 cm"; however, the physical drivers of evaporation can differ between the ground and shrub canopy despite modest differences in elevation. (d) Thanks! The same as the previous question, we agree with what you said. We proposed this assume just think it could be better than the

original assume which ignoring the ground evaporation. The ground of the shrub patch intervals is covered by herbs, and the herbs also intercepts precipitation. The herbs coverage is much higher than the shrub coverage. As the height of the shrub is only about 35 cm, the ground evaporation could not be ignored. In a similar ecosystem, Zheng (2015) reported that in the growing season, the mean evaporation rates for the shrub and nearby grass land were 2.80 mm d-1 and 2.52 mm d-1, respectively. The results of Zheng (2015) showed that the evaporation rates of the two parts can be roughly equal and the grass land evaporation should not be neglected.

9) The manuscript is in need of significant English language editing. As it would take too much time to identify and suggest changes for all of the necessary language editing, an example in each section are provided to guide the authors during their revisions: 10) Thanks! The English will be improved throughout the text by an English language editing company before it is resubmitted.

Abstract, P1, L10– "has not get enough attention" should be "has not gotten enough attention". . . but, my opinion is that the language shouldn't be so colloquial. It would be better to state something like "has not received enough attention"

Thanks! It has been modified as "has not received enough attention".

Introduction P2, L11– "The gross precipitation reaches the canopy. . ." should be "The gross precipitation that reaches the canopy. . ." and the authors incorrectly state that the canopy partitions precipitation only into interception, stemflow and "free" throughfall. This ignores all "release" throughfall produced from canopy contact. Why not just say "interception stemflow and throughfall" as the general term "throughfall" implies the sum of free and release throughfall?

Thanks. We have modified the sentence according to your suggestion.

Methods P6, L7– "rainfall events were discretized by assuming without rainfall between events of 12 h. . ." should be something like "rainfall events were discretized by assum-

ing a minimum inter-event time of 12h. . ." Minimum inter-event time is a common term in precipitation partitioning literature (i.e., Dunkerley, 2015, Hydrol Process, 29, 3294 and Llorens et al., 2014, J Hydrol, 512, 254). I would also recommend the authors' follow the convention for introducing species: latin name (taxonomic authority, common name). Thereafter, the use of the abbreviated latin name is typically used.

Thanks a lot. The sentence "rainfall events were discretized by assuming without rainfall between events of 12 h. . ." was modified as your suggestion.

Results P15, L13 – I think the line "which occupying 14.4% of the total observed interception" should be "which accounted for 14.4% of the total observed interception"

Thanks. We have modified the sentence according to your suggestion.

Discussion P17, L11-12 – "The proportion pt also has important in the stemflow of course" needs to be rewritten for clarity as I'm unsure what the authors are saying.

It is rewritten as "The proportion pt also has important influence in the generation of stemflow of course, as it determines the percentage of interception that converts to stemflow."

Conclusion P26, L9 to P27, L2 – this statement is unclear. Does "available water that free fell and drained along the stem" mean "stemflow"? Why would "free" falling droplets "drain along the stem"? Or, does this statement simply mean "throughfall and stemflow"? Please revise for clarity.

Thanks! The sentence was revised as: 21.44% of gross rainfall was intercepted by the canopy, throughfall and stemflow, accounting for 29.26% and 49.30% of gross rainfall, respectively, were available water that reached the soil ground during the growing season of 2012.

References

David D.: A new method for determining the throughfall fraction and throughfall depth

in vegetation canopies, J. Hydrol., 385(1-4), 65-75, 2010.

Macinnis-Ng C. M. O., Flores E. E., Müller H., Schwendenmann L.: Throughfall and stemflow vary seasonally in different land-use types in a lower montane tropical region of panama, Hydrol. Process., 28(4), 2174-2184, 2014.

Muzylo A., Llorens P., Valente F., Keizer J. J., Domingo F., Gash J. H. C.: A review of rainfall interception modelling, J. Hydrol., 370(1-4), 191-206, 2009.

Muzylo A., Valente F., Domingo F., Llorens P.: Modelling rainfall partitioning with sparse gash and rutter models in a downy oak stand in leafed and leafless periods, Hydrol. Process., 26(21), 3161-3173, 2012.

Návar J.: The performance of the reformulated gash's interception loss model in mexico's northeastern temperate forests, Hydrol. Process., 27(11), 1626-1633, 2013.

Zheng X.: Partitioning evapotranspiration in shrub-encroached grassland in Inner Mongolia: Model simulation and application, Beijing Normal University, 2015 (in Chinese with English abstract).
* * *
[Figure]

[Figure]

**Fig. 1.** Figure 1 Stemflow collection apparatus on a branch

---

## Author Comment (AC2) · 4 Apr 2017

Dear anonymous Referee #2,

We would like to thank you for your valuable and constructive comments. The comments are very helpful to the improvement of the manuscript, and will be well incorporated into the revision of the paper. The following paragraphs respond to your comments one by one.

General comments:

The manuscript (ms) reports on measurement and modelling of rainfall interception by a deciduous shrub species. Although several studies have already been published on

the rainfall interception by deciduous shrubs, only in a few modelling was done. The specific characteristics of these cover-types, with drastic seasonal changes in canopy structure, could make this study quite useful and liable to provide relevant contributions on the subject. However, I think that the ms has several important shortcomings in the present form and that its focus/rationale needs to be improved and clarified. In my opinion, the ms needs a major revision before it can be considered for publication in HESS.

Thank you very much. We appreciate your suggestions and revised the ms accordingly, and improve and clarify its focus and rationale. Hope it can reach the standard of HESS.

Specific comments:

1. The English is poor and the ms does not read well (sometimes it is hard to understand what the authors are trying to say).

Thanks! The revised version will be edited and refined by a company of language services before it is resubmitted.

2. In some cases, standard terminology on rainfall interception is not used correctly by the authors. Usually, "interception" is used to describe the interaction process between rainfall and vegetation while "interception loss" refers to its evaporation component (the amount of water retained in plants surfaces that evaporates back into the atmosphere). The authors use the term "interception" with both meanings resulting in a confusing text (e.g., page 2, line 10−12, "The gross precipitation reaches the canopy is redistributed to interception, stemflow and free throughfall"; page 18, lines 12−14, "The stemflow are the part of interception that run down the stem, so if the interception reduces, the stemflow would reduce"). The authors should check all text and differentiate between concepts using the appropriate terminology.

Thanks! We checked text throughout and made sure the appropriate terminology was

used. The details will be showed in the revised ms

3. The description of the experimental site and vegetation characteristics needs further information and to be reorganized. In page 5, lines 16−17, the authors say "The coverage of shrub is 26%, and the height of shrub is 35.4 cm". How was this cover fraction evaluated? Does this value correspondent to the (average?) cover fraction of an individual plant or to the total percentage of cover area in the experimental site? Is the given value for shrub height a mean? What about other characteristics of individual plants (average number of stems per plant, mean diameter of each stem,...)? Although some of this data is presented in the ms, it is dispersed across several sub-sections (e.g., page 13, line 7). All this disperse information should be gathered together.

Thanks! More details were added in the ms.

Three 5 m by 5 m sample areas were selected randomly in the experimental site to survey the plant characteristic. evaluated by measuring each shrub patches area in three 5 m by 5 m sample areas. The shrub patches were treated as ellipses, and their axes were measured using a measuring tape. And the height of the shrub was measured in each patch. Six patches were selected to count their stems and measure their base diameters using a vernier caliper. The aboveground biomass of three of the six patches was collected and oven-dried at 65 °C and then weighed in June and another 3 patches in August.

The stem coverage (cs) within the shrub patches was estimated by taking and interpreting photos above the canopy in the leafless period. In the leafed period, the leaves in shrub patches nearby the stemflow and throughfall measuring plots were collected and scanned to calculate the one-sided leaf area before they were oven dried at 65 °C and weighted. The area of the shrub patches was measured and the leaf area index (LAI) was calculated as the leaf area of per unit of shrub patch area. The relationship between leaf coverage (cl) and LAI was simplified as linear and was set as 1 when the LAI was maximum (eq. (1)).

cl=LAI/LAImax (1)

The LAI was thought to have linear relationship with day of year in the foliation and defoliation period and the linear equations were built up by measuring the LAI at the same time of the stemflow measurement in 11, 20 and 29 June, 17 and 31 July, 22 August and 2 and 11 September, respectively.

The stem density of the P. fruticosa is 385 stems per m2 in the shrub patches. The given shrub height is a mean. The plant characteristic will be gathered together in Results 3.1.

4. Concerning the measurement of rainfall, throughfall, stemflow and micrometeo-rological variables, important information is missing. Location and type of the different gauges (tipping bucket and/or bottles) are not given. How were rain gauge locations chosen? How far from the edge of the patches were they placed? Did the gauges/bottles stay in fixed positions or were moved to new random positions each time they were measured? At what height were the micrometeorological sensors installed? Where were these sensors installed: above a shrub patch or in open areas between patches? What is the footprint for these data? Although micrometeorological data is from a previous study, it should be briefly described here. All this information is relevant to the study (measurement and modelling of rainfall interception) and should be presented in the ms. An aerial photography of the site with the location of the used devices (rainfall, throughfall and stemflow gauges and the Bowen ratio tower), would be helpful.

Thanks! The gross precipitation was measured by a tipping rainfall gauge (ARG100, Campbell, USA, 0.2 mm per tip). The rainfall gauge was located in a relatively flat, open area, and about 1 m above the ground, much higher than the shrubs. The throughfall and stemflow were collected by bottles. The gauge and bottles stayed in fixed positions in order not to disturb the shrub patches too often. The distance between the rainfall gauge and stemflow bottles / throughfall bottles was less than 100 m.

The Bowen ratio tower is located on a shrub patch. The net radiation was measured at 2 m above the canopy with a 240-100 net radiation sensor (NOVALYNX, USA). The air temperature and relative humidity were measured at 1m and 2 m above the canopy with the 225-HMP50YA sensor (NOVALYNX, USA). The soil heat flux was measured using a HFP01 sensor (Dynamax Inc., USA) at a depth of 0.05 m underground. Soil temperature and moisture was measured using a ECH2O 5TE sensor (Decagon Devices, USA) at a depth of 0.1 m underground. Wind speed and direction were measured using a 05103-5 sensor (RM-YOUNG, USA) at 2 m above the canopy. The footprint was not accurately analyzed. However, the Bowen ratio tower located in the core of a nearly homogeneous area larger than 500 m in diameter, which means its footprint can be consistent regardless of the wind direction.

No aerial photography of the site was taken. The Bowen ration tower was shown in Figure 1, and the stemflow and throughfall were measured in the vicinity of 100 meters.

<Figure 1 The Bowen ration tower>

5. To extrapolate stemflow measurements to the total patch area the authors used a stepwise methodology to derive a regression model. Which were the independent variables considered in this analysis? Though the final model has only three variables (page 7, eq. 1), were other structural features/rainfall characteristic considered? One of the variables included in eq. 1 is q, "the number of rainfall events that generate stemflow" (page 7, lines 5−6). How was q evaluated? In page 6, lines 14−16, it is stated that "Because it is very difficult to collect stemflow in the remote area, we did not measure stemflow for each rainfall events, and we measured and recorded stemflow eight times during the study period". Given this, how do the authors know the number of rainfall events that generate stemflow in each period?

The considered independent variables in the regression modelling included stem canopy structure parameters (basal diameter, basal area, stem length, stem biomass, leaf biomass, leaf area, aboveground biomass) and rainfall characteristics (rainfall

amount , rainfall intensity, maximum rainfall intensity in 10 min, rainfall duration, wind speed). The final model with three variables entered is the best model with the least root-mean-square of residuals.

q is the number of rainfall events that generate stemflow, that is, the amount of rainfall larger than that required to saturate the stem ($P\_G\hat{}$"), which can be calculated using the micrometeorological and canopy data.

6. It seems to me that the authors do not totally understand the sparse version of Gash's analytical model.

(a)They say that the model requires several parameters and refer that "the free throughfall coefficient (p) and the canopy coverage (c)" are two of them (page 9, line 19). In page 18, lines 16−17, they restate that p is a parameter of the model. This is not correct! The sparse version of the Gash model only requires c, the proportion of covered area relative to the total area.

Thank! We agree with you, and modified the description in the ms. As a substitute for direct measurement, the coverage can be assumed to be one minus free throughfall coefficient (Shi et al., 2010; Fan et al., 2014). The free throughfall coefficient can be estimated as the slope of the linear regression of throughfall against gross precipitation for small rainfall events that were insufficient to exceed canopy storage capacity (Jackson, 1975; Shi et al., 2010; Fan et al., 2014).

(b)Although not acknowledged, the authors mix the sparse version of the Gash model proposed by Gash et al. (1995) with the slightly different version presented later by Valente et al. (1997) (e.g., the amount of rainfall required to saturate the trunks (Pg ") is only defined by Valente et al. (1997)).

Thanks! We have modified the description in the ms. In the sparse version of the Gash model proposed by Gash et al (1995), the amount of rainfall required to saturate the trunks was defined as St/pt.

**HESSD**

(c)Two of the most important parameters of the sparse version of the Gash model are R and Ec (and not E, as it is said in page 10, line 3). According to Gash et al. (1995), these parameters are the mean rainfall rate and the mean evaporation rate during saturated conditions, respectively, and should be representative for the whole modelling period. Following Gash (1979), the method usually used to derive R is the average of all hours with rainfall equal or greater than 0.5 mm (two bucket tips) for the whole modelling period. How did the authors calculate R? Nothing is said about this. The same happens with Ec. The authors say they used data obtained with the Bowen Ratio/Energy Balance method (BREB) (page 11, lines 10−11), but do not say how.

Thanks! For the sparse version of the Gash model, the mean rainfall rate (R ÌĚ) was the average of all hours with rainfall equal or greater than 0.4 mm (two tips) for the whole modelling period. For the new version model, the (R_j ) ÌĚ was the average of all hours in each rainfall event (j).

The E estimated using BREB was 10 min interval. For the sparse version of the Gash model, the mean evaporation (E ÌĚ) was the average of all hours with rainfall equal or greater than two tips for the whole modelling period. For the new version model, the (E_j ) ÌĚ was the average of all hours in each rainfall event (j).

These details have been add to the ms.

(d)Besides, it seems that the authors do not fully understand the meaning of Ec. It represents the evaporation rate at which intercepted water can evaporate from a fully saturated canopy. But the authors say that Ec = E/c (page 10, line 8). What is the meaning of E in the context of the sparse version of the model? If E is the actual measured evaporation rate from a fully wet vegetation and it is assumed that the only water source is the studied wet vegetation then this relationship is correct. Otherwise, it is not. It seems to me that the authors did not get it correctly. In fact, the authors say (page 23, lines 12−13) that "the average evapotranspiration in P. fruticosa shrub meadow was 0.11 mm h−1 during the experimental period". How was this calculated?

[Figure]

They also refer that "the hourly evaporation varied greatly in different time, ranging from −0.04 to 0.87 mm h−1, controlled mainly by radiation" (page 23, lines 16−17). However, during rainy/cloudy conditions (when the canopy is saturated), radiation is typically low and evaporation rate should not change much. This may suggest that the aforementioned values include periods where the vegetation is not fully wet, possibly not representative of saturated canopy conditions.

Thanks! E is the actual measured evaporation rate from a fully wet vegetation. E ÌĚ = 0.11 mm h−1 was the average actual measured evaporation rate of all hours with rainfall equal or greater than 0.4 mm (two tips) for the whole modelling period.

It is true that during rainy/cloudy conditions, radiation is typically low. However, the weather is changeable, especially on the Plateau and in summer. It is common that there are a few hours of sunshine between two showers, or alternation of sunny day and shower with inter-event times less than 8 h. In this condition, the radiation and evaporation can change greatly and have a high peak in the fine weather (see the grey parts in Figure 2).

<Figure 2>

(e)The authors present three equations (page 10) to calculate the different components of rainfall interception (interception loss, stemflow and throughfall). Although based in the model version proposed by Valente et al. (1997) (again not acknowledge here), these equations do not describe the sparse version of the Gash model. As the authors say (page 9, lines 11−13), one of the assumptions of the model is that Ec and R are assumed constant over the whole modelling period. However, while gross precipitation seems to be constant (since the j index is missing in Pg ), but should not, Ec and R can change from storm to storm (because they have a j index). Moreover and contrary to the current practice, trunk storage capacity (St) is expressed in mm on a projected cover area basis (that is why it is necessary to multiply St by c in eq. 4 and 5). Whenever the units of a parameter are water depth (e.g., mm), it should be clearly stated in

the text what is the reference area (e.g., ground area, covered area, . . .).

Equations you referred to have been modified to adapt the sparse version of the revised Gash model. These equations were all right when used in the modelling, however, when wrote the ms, they were copied from the equations for the new version model, and some details were not modified. These mistakes were corrected. All the symbols in text and eqs. were checked and mistakes were revised.

The patch canopy water storage capacity (S, mm), the leaf water storage (Sl, mm) and the stem water storage (Ss, mm) were defined as the water storage per patch area and were estimated using the data of patch aboveground biomass, stem biomass and leaves biomass along with the patch area from plant survey. The Sl would change along with the foliation and defoliation. The Ss was thought to be constant in the experimental period as the stem grows quite slowly in the cold region.

(f)The authors present a new version of this model to adapt it to the studied deciduous shrub (page 10, line 17 to page 12, line 10). They assume that the evaporation rates from all the vegetation components (canopy, stems and inter-patch herbs) are the same. I am not sure if this is a realist assumption, since roughness and/or the micrometeorological conditions are seldom similar.

Thanks! We agree to the fact that there is difference between the evaporation rates from different vegetation components due to the reasons the reviewer identified. We proposed this assume just think it could be better than the original assumes considering the leafless period and the height of the shrub. Zheng (2015) reported that in the growing season, the mean evaporation rates for the shrub and nearby grass land were 2.80 mm d-1 and 2.52 mm d-1, respectively. The results of Zheng (2015) showed that the evaporation rates from shrub patches and inter-patch grass land can be roughly equal and the grass land evaporation should not be neglected. The original model assumed the trunk evaporation only happens in the drying out period and the inter-patch evaporation is assumed as zero. The original assume of course is not the real fact,

too. Especially, in the leafless period, rain can fall on the stems and then evaporated directly. The evaporation is a complicated process, and it is difficult to distinguish evaporation rates on different surfaces. The original and present assumes are all for simplify in modeling these process.

(g)Nevertheless, the requirements of the energy and water balances should be met. When all the vegetation is saturated, the measured BREB values (E) represent the evaporation of the total area and not just of the wet shrubs cover (see my previous comment 6.(d)). It seems to me that the authors did not took into account the water balance equation in their new modelling proposal (page 12, eqs. 8 and 9 and Table 2). How were these new equations obtained? An explanation is needed.

Thanks! When all the vegetation is saturated, the measured BREB values (E) represent the evaporation of the total area and not just of the wet shrubs covered. And it is assumed that the evaporation rates from all the vegetation components (canopy, stems and inter-patch herbs) are the same. So, the BREB values (E) also represent the evaporation rates of the canopy.

The new equations for the new model were obtained as following Supplement I:

(h)Another important missing information is the "time-step" used to run the model. Although the model is storm-based, it is usually run assuming that each rain day is an independent rainfall event. Which procedure did the authors used?

The micrometeorological data is 10 min average, and the evaporation was calculated in 10-min step firstly. For the new version model, the 10-min interval rainfall intensity and evaporation rate were averaged in each rainfall event. The model was run basing on storm record, not basing on rain day. A simple Matlab (Version R2008b) procedure was wrote by the authors and was used in the modeling.

7. The authors present results on the water storage capacity of leaves and stems (page 13, lines 12−16) but do not explain how they were obtained. Only the method used

to measure branch water storage capacity is described. Further- more, they do not explain how ml were converted into mm (page 13, line 18). What is the reference area in the latter? The method used to estimate another model parameter (pt) is not also described in the text.

The canopy storage capacity was measured under artificial simulated rainfall. Firstly, the stretch angle of branches of P. fruticosa were measured in situ. Then, they were excavated and carefully took back to the laboratory in a whole plant with some soil to assure that they were fresh. In the laboratory, branches were cut off from the base and weighed and then fixed on a wood base at their original angle. Artificial simu- lated rainfall was implemented immediately. Other three fresh bare stem without leaves were also experienced in the artificial simulated rainfall. After rainfall, each branch was weighed again, the difference of the weight before and after the rainfall was the water the branch stored. Then, leaves were picked away, the bare stems were dried to their original weight (when the weight of three bare stems equaled their original weight). The simulated rainfall was implemented again. After rainfall, each stem was weighed again, the difference of the weight before and after the rainfall was the water the stem stored. The difference of the stored water by the branches and the stems is the water stored by the leaves. The leaves were scanned with a scanner to calculate the one-sided leaf area. The stems and leaves then were oven-dried and weighed. Totally, 33 branches were measured in the simulated rainfall at a rainfall intensity of 10.9 mm h-1 and a rainfall duration of 1h. 10.9 mm h-1 is the minimum intensity that the rainfall simulator could reach. The relationship between stem water storage capacity (Cst, ml) and stem dry mass (Mst, g), and leaves water storage capacity (Clf, ml) and leaves dry mass (Mlf, g) were thought to be linear. Wb (ml g-1), Wst (ml g-1) and Wlf (ml g-1) were the branch, stem and leaf water storage capacity per mass, respectively.

$$Wst = Cst/Mst \quad (3)$$

$$Wlf = Clf/Mlf \quad (4)$$

The patch canopy water storage capacity (S, mm), the leaf water storage (Sl, mm) and the stem water storage (Sst, mm) were defined as the water storage per patch area and were estimated using the data of patch aboveground biomass, stem biomass and leaves biomass along with the patch area from plant survey.

Sst=0.1*Mst*Pst/A (5)

Sl=0.1*Ml*Pl/A (6)

where Pb, Pst and Pl are the aboveground dry biomass (g), stem dry mass (g), leaf dry mass (g) of a shrub patch; A is the patch area (cm2). The leaf mass and Sl were thought to have linear relationship with day of year along with the foliation and defoliation. The Sst was thought to be constant in the experimental period as the stem grows quite slowly in the cold region.

pt is estimate of the slope of the linear regression of stemflow against PG.

8. Considering the characteristics of the studied vegetation (deciduous), it would be expectable the presentation of data on the time evolution of some parameters, namely canopy cover, and canopy and stem storage capacities. This would provide support on the need of using time variable parameters instead of the usual constant values. Besides, as the authors used different Ec and R, it would be relevant to have a graph of their values along the modelling period. Neither of these variable parameters, nor the constant ones needed to run the sparse version of the Gash model are given in the ms.

Good idea. The coverage and water storage capacity of stems were thought to be constant as the stems grow very slowly in the high and cold region.

The time evolution of leaf area index (LAI) was showed in Figure 3. The LAI increased linearly until the end of July, and then decreased linearly until the end of experiment. The LAI had a maximum of 2.47 measured in 31 July, 2012. The leaf storage capacity also change along with the LAI with a maximum of 0.59 mm (Figure 4). The coverage

of leaf ranged 0.53 to 1 and the total coverage ranged 0.84 to 1 (Figure 5). The rain intensity (rain larger than one tip) and the mean evaporation rates during the rain event were showed in Figure 6. The rain intensity ranged 0.06 to 2.40 mm h-1 with an average of 0.76. The mean evaporation rates ranged 0.03 to 0.17 mm h-1 with an average of 0.11 mm h-1.

<Figure 3 The change of leaf area index (LAI) along with the day of year (DOY)>

<Figure 4 The change of leaf storage capacity (Sl) along with the day of year (DOY)>

<Figure 5 The change of patch coverage (C) and leaf coverage (Cl) along with the day of year (DOY)>

<Figure 6 The intensity of rain events and the evaporation rates during the storms. The horizontal axis is the day of year (DOY) when the rains began. (Rain events which was only one tip were not showed)>

9. The performance of the tested models was only evaluated by the total error (EE). However, EE per se does not evaluate the quality of model performance throughout the simulation period. For that purpose, authors should have applied some additional measure, such as modelling efficiency (see Mayer and Butler, 1993, Ecol. Modelling, 68: 21-32).

According to the method of Mayer and Butler (1993) , the modelling efficiency was calculated for the two models (Table 1). The variable parameters Gash model had better performance than the revised Gash model in all three rainfall partitioning. The modelling efficiency of throughfall, stemflow and interception of the VPG were 0.99, 0.99 and 0.79, respectively.

Table 1 Validation measures for throughfall, stemflow and interception of the revised Gash model (RG) and the variable parameters Gash model (VPG). MAE: the mean absolute error; MA%E: mean absolute percent error; RMSE: the root mean square error; RMSE%: the ratio of the EMSE to the range of observed values; EF: modelling

efficiency. Throughfall Stemflow Interception RG VPG RG VPG RG VPG MAE 7.47 1.52 6.56 2.37 3.58 2.17 MA%E 39.09 10.52 15.08 11.00 23.02 14.75 RMSE 10.31 2.04 11.99 3.27 4.05 2.72 RMSE% 19.20 3.80 10.55 2.88 22.84 15.38 EF 0.64 0.99 0.89 0.99 0.53 0.79

10. As in many other studies, the authors have conducted a sensibility analysis of the sparse version of the Gash model. The question is: what is new about this? If they have used their own model version this could be interesting. What has been done is just a repetition that does not bring any new insight on the subject. Furthermore, the presentation of the results and their discussion are incomplete. Why is not shown a positive change of c in the graphs (Fig. 2)?

In what concerns canopy cover (S), model sensitivity to this parameter was found to be very small which is not in accordance with most previous findings. However, the authors state that "the results in this paper are in accordance with [the] results" of other studies and will not be discussed in the ms (page 17, lines 6−8). On the other hand, they state that "the canopy storage capacity is the most important parameter in the interception modelling" (page 19, lines 11−12) which is contradictory. In my opinion, the authors should focus their work in what is new and relevant to the subject (modelling the rainfall interception process in a deciduous shrub cover).

Thanks. The sensibility analysis will be deleted in the revised ms, and the contradictory description will be eliminated.

11. Minor comments:

(a)Page 3, line 13 & page 4, line 6 − replace "Analytical" by "Conceptual". The Rutter model is not an "analytical model".

We replaced "Analytical" with "Conceptual".

(b)Page 4, line 6 − what are semi-constants?

There are two values for a parameter, such as a $S_c$ for leafless period and another for
leafed period. Maybe they can be said constants after all.

(c)Page 4, lines 16−20 & page 4, lines 1−6 − the objectives of the work should be presented in a concise way. This text should be simplified and avoid repetitions.

Thanks! The objectives of this study were (1) to measure and analyze the rainfall interception, stemflow and throughfall of P. fruticosa shrub on the Qinghai-Tibet Plateau, China, and (2) to adapt the revised Gash model to the deciduous shrub using the directly measured variable parameters and to compare the results. The adapted model will consider the special canopy structure of deciduous shrubs. Some hypotheses would be reset due to the special canopy structure and weather condition. The changes of canopy parameter relating to the process of foliation and defoliation were monitored and some important canopy parameter measured directly.

(d)Page 6, line 6 − specify tip sensitivity of rainfall gauge.

The sensitivity is 0.2 mm per tip. We added this information to the ms.

(e)Page 6, lines 16−17 − there are only seven periods with measurements.

Thanks. We measured 8 times. Unfortunately, some stemflow and throughfall data was missed in July 17, 2012. But the data of leaf area index and rainfall characteristic in this period were analyzed. We added more details in the ms to illustrate it.

(f)Data from the 17th July 2012 is missing (Tables 3, 4 and 5). Authors should mention that in the text.

We added more details in the ms to illustrate it.

(g)Page 6, line 18 − are stem diameter units correct (mm)? A stem with 3.4 mm seems too small to support any collecting device to measure stemflow.

Yes. It is hard to collect the stemflow from such small stem. But we tried and found a device (Figure 7) to do it and it worked well. The stemflow was measured following the method of Zhang et al. (2015). A small sink was established by wrapping a piece of

aluminium foil at the base of stem, the collected stemflow in the sink was drained to a storage bottle through a flexible plastic tube. The sink edges were 1~2 mm away from the wrapped shrub stems.

<Figure 7 Stemflow collection apparatus on a branch (photo courtesy of Si-Yi Zhang)>

(h)Page 7, line 19 − what is the meaning of "10 min frequency data"? Do the authors mean "10 min average data"?

Yes. It means "10 min average data" and was modified in the revised ms.

(i)Page 8, lines 15−17 − this sentence should go to the discussion section.

We'll discussed it in the discussion section of the revised ms.

(j)Page 9, line 6 − replace (Gash, 1975) by (Gash, 1979).

We revised it as your suggestion.

(k)Page 11, line 10 − the acronym BREB should be previously defined.

We'll defined it in the Section Materials and methods.

(l)Page 12, lines 1 and 4 − the subscript j is missing in the symbols.

Thanks. All the symbols in text and eqs. were checked and mistakes were revised.

(m)Page 12, line 15 − do the authors mean a storm with 50 years' return period?

Yes. The sentence was revised as: "The total rainfall amounted to 531.0 mm, and ranged from 0.2 mm and 40.0 mm except for a 106.2 mm storm with 50 years' return period in August."

(n)Page 13, line 9 − according to eqs. 1 and 10, symbol for stemflow should be $SF_v$ , not $SF_b$.

Thanks. All the symbols in text and eqs. were checked and mistakes were revised.

(o)Table 1 − please remove the reference to Pg ; this variable is not in table.

It is removed.

(p)Table 3 − table not referred in text.

We now refer it in Section 3.2 Observed rainfall partitioning pattern.

(q)Figure 1 b) − I do not understand this graph. What do the authors want to show with it? Please explain.

Yes. It is somewhat hard to understand. It showed the cumulative number and amount of rainfalls whose rainfall depths were not larger than a given rainfall depth. We delete it and add a figure about the intensity and evaporation rate.

References

Fan J., Oestergaard K. T., Guyot A., Lockington D. A.: Measuring and modeling rainfall interception losses by a native banksia woodland and an exotic pine plantation in subtropical coastal australia, J. Hydrol., 515, 156-165, 2014.

Jackson I. J.: Relationships between rainfall parameters and interception by tropical forest, J. Hydrol., 24(3), 215-238, 1975.

Mayer D. G., Butler D. G.: Statistical validation, Ecol. Model., 68(1–2), 21-32, 1993.

Shi Z., Wang Y., Xu L., Xiong W., Yu P., Gao J., Zhang L.: Fraction of incident rainfall within the canopy of a pure stand of pinus armandii with revised gash model in the Liupan mountains of china, J. Hydrol., 385(1-4), 44-50, 2010.

Zhang S., Li X., Li L., Huang Y., Zhao G., Chen H.: The measurement and modelling of stemflow in an alpine myricaria squamosa community, Hydrol. Process., 29(6), 889-899, 2015.

Zheng X.: Partitioning evapotranspiration in shrub-encroached grassland in Inner Mongolia: Model simulation and application, Beijing Normal University, 2015 (in

Chinese with English abstract).

Please also note the supplement to this comment:
http://www.hydrol-earth-syst-sci-discuss.net/hess-2016-589/hess-2016-589-AC2-supplement.pdf

[Figure]

**Fig. 1.** The Bowen ration tower

[Figure]

**Fig. 2.** Rain pulses and fluctuation of net radiation and evaporation during June 16, 2012 and June 22, 2012

[Figure]

The change of leaf area index (LAI) along with the day of year (DOY)

$LAI=7.43-1.88\times10^{-2}DOY\ (212<DOY<273)$
$R_a^2=0.857, p=0.049$

$LAI=-3.43+3.25\times10^{-2}DOY\ (151<DOY\leq212)$
$R_a^2=0.965, p=0.002$

**Fig. 3.** The change of leaf area index (LAI) along with the day of year (DOY)

$S_l=1.59-4.02\times10^{-3}DOY\ (212<DOY<273)$

$R_a^2=0.857,\ p=0.049$

$S_l=-0.74+6.97\times10^{-3}DOY\ (151<DOY\leq212)$

$R_a^2=0.965,\ p=0.002$

**Fig. 4.** The change of leaf storage capacity (Sl) along with the day of year (DOY)

The plot shows Coverage vs DOY with two series: $C_l$ (filled circles) and $C$ (filled triangles).

**Fig. 5.** The change of patch coverage (C) and leaf coverage (Cl) along with the day of year (DOY)

[Figure]

**Fig. 6.** The intensity of rain events and the evaporation rates during the storms. The horizontal axis is the day of year (DOY) when the rains began. (Rain events which was only one tip were not showed)

[Figure]

**Fig. 7.** Stemflow collection apparatus on a branch (photo courtesy of Si-Yi Zhang)

**Supplement:**

**Supplement I**

The water income of stems includes: (1) the direct rain falling on the stems, $(1-c_{l,j})c_s P_{G,j}$, (2) the water draining from leaf canopy, $p_t c_{l,j}(P_{G,j}-P'_{G,j})(1-\overline{E_j}/\overline{R_j})$. The water outcome of the stems before stemflow generation is the stems (not shaded by leaves) evaporation, $(1-c_{l,j})c_s\overline{E_j}P_{G,j}/\overline{R_j}$. The evaporation from stems shaded by the leaf were ignored. The water balance of stem when the stemflow generation is:

$$(1-c_{l,j})c_s P_{G,j}+p_t c_{l,j}(P_{G,j}-P'_{G,j})(1-\overline{E_j}/\overline{R_j})-(1-c_{l,j})c_s\overline{E_j}P_{G,j}/\overline{R_j}=S_s \qquad (1)$$

The magnitude of rainfall required to saturate the stem ($P''_G$) is the solve of the $P_{G,j}$ in eq. (2):

$$P''_{G,j}=(S_s\overline{R_j}/(\overline{R_j}\text{-}\overline{E_j})+c_{l,j}P'_{G,j})/(c_s+p_t c_{l,j}-c_s c_{l,j}) \qquad (2)$$

Considering the assumption that the stem evaporation happens in the whole rainfall period but not only in the drying out period, the evaporation from stem in $n–q$ storms insufficient to saturate the stem ($P_{G,j}<P''_{G,j}$) includes: (1) the direct rain falling on the stem, $(1-c_{l,j})c_s P_{G,j}$, (2) the water draining from leaf canopy, $p_t c_{l,j}(P_{G,j}-P'_{G,j})(1-\overline{E_j}/\overline{R_j})$; the evaporation from stem in $q$ storms sufficient to saturate the stem ($P_{G,j}\geq P''_{G,j}$) includes: (1) stem evaporation during the rain period, $\sum_{j=1}^{q}\overline{E_j}c_s(1-c_{l,j})P_{G,j}/\overline{R_j}$, and (2) the stem water storage capacity $qS_s$.

The stemflow could be recalculated by following equation:

$$\sum_{j=1}^{q}SF_j=\sum_{j=1}^{q}p_t(1-\overline{R_j}/\overline{E_j})(c_{l,j}(P_{G,j}-P''_{G,j})+c_s(1-c_{l,j})(P_{G,j}-P''_{G,j})) \qquad (3)$$

The stemflow includes two parts: (1) the rain drains from leaf canopy after the stem is saturated with a ratio of $p_t$, $\sum_{j=1}^{q}p_t(1-\overline{R_j}/\overline{E_j})c_{l,j}(P_{G,j}-P''_{G,j})$, and (2) the rain fall on the stem directly, and converts to stemflow at a ratio of $p_t$, $\sum_{j=1}^{q}p_t(1-\overline{R_j}/\overline{E_j})c_s(1-c_{l,j})(P_{G,j}-P''_{G,j})$.

**Abbreviations**:

$p_t$: stemflow ratio that rainfall is diverted to the trunks;

$c_{l,j}$: leaf coverage in storm $j$;

$c_s$: stem coverage;

$P_{G,j}$: rainfall depth of storm $j$, mm;

$P'_{G,j}$ : the amount of rainfall required to saturate the leaf canopy, mm;

$\overline{E_j}$ : the evaporation rate during storm j, mm h$^{-1}$;

$\overline{R_j}$ : the rainfall intensity during storm j, mm h$^{-1}$;

$S_s$: stem water storage capacity, mm;

$P''_G$ : the magnitude of rainfall required to saturate the stem, mm;

$q$: The number of rains which generated stemflow;